



**Reviews and syntheses: Use and misuse of peak intensities from**
**high resolution mass spectrometry in organic matter studies:**
**opportunities for robust usage**
William Kew[1], Allison Myers-Pigg[2], Christine H. Chang[2], Sean M. Colby[2], Josie Eder[1], Malak
M. Tfaily[3], Jeffrey Hawkes[4], Rosalie K. Chu[1], James C. Stegen[2*]
[1]Environmental Molecular Sciences Laboratory, Richland, WA 99352, USA
[2]Pacific Northwest National Laboratory, Richland, WA 99352, USA
[3]Department of Environmental Science, University of Arizona, Tucson, AZ, 85719, USA
[4]Department of Chemistry, University of Uppsala, Uppsala, 75124, Sweden
*Correspondence to: James C. Stegen (James.Stegen@pnnl.gov)
**Abstract**
Earth's biogeochemical cycles are intimately tied to the biotic and abiotic processing of organic matter (OM).
Spatial and temporal variation in OM chemistry is often studied using high resolution mass spectrometry (HRMS).
An increasingly common approach is to use ecological metrics (e.g., within-sample diversity) to summarize high-
dimensional HRMS data, notably Fourier transform ion cyclotron resonance MS (FTICR MS). However, problems
arise when HRMS peak intensity data are used in a way that is analogous to abundances in ecological analyses (e.g.,
species abundance distributions). Using peak intensity data in this way requires the assumption that intensities act as
direct proxies for concentrations, which is often invalid. Here we discuss theoretical expectations and provide
empirical evidence why concentrations do not map to HRMS peak intensities. The theory and data show that
comparisons of the same peak across samples (within-peak) may carry information regarding variation in relative
concentration, but comparing different peaks (between-peak) within or between samples does not. We further
developed a simulation model to study the quantitative implications of both within-peak and between-peak errors
that decouple concentration from intensity. These implications are studied in terms of commonly used ecological
metrics that quantify different aspects of diversity and functional trait values. We show that despite the poor
linkages between concentration and intensity, the ecological metrics often perform well in terms of providing robust
qualitative inferences and sometimes quantitatively-accurate estimates of diversity and trait values. We conclude
with recommendations for using peak intensities in an informed and robust way for natural organic matter studies. A
primary recommendation is the use and extension of the simulation model to provide objective, quantitative
guidance on the degree to which conceptual and quantitative inferences can be made for a given analysis of a given
dataset. Without objective guidance, researchers that use peak intensities are doing so with unknown levels of
uncertainty and bias, potentially leading to spurious scientific outcomes.



## 1 Introduction

Organic matter (OM) plays a central role in Earth's biogeochemical cycles, and is both a resource for and product of metabolism. The detailed chemistry of OM (e.g., nominal oxidation state) can modulate and reflect biogeochemical rates and fluxes within and across ecosystems (e.g., LaRowe and Van Cappellen, 2011; Boye et al., 2017; Garayburu-Caruso et al., 2020), yet our understanding of this complexity is limited by our analytical abilities to view it (Steen et al., 2020; Hedges et al., 2000; Hawkes and Kew, 2020a). Given the importance of OM chemistry to biogeochemical cycling, there is a need to understand how and why that chemistry varies through space and time. To help meet this need, there has been growing interest in using concepts and methods from ecology to study the chemogeography and chemodiversity of OM in a variety of ecosystems (e.g., Kujawinski et al., 2009; Kellerman et al., 2014; Tanentzap et al., 2019; Danczak et al., 2021). This is a promising approach as there are many conceptual parallels between the chemical species that comprise OM and the biological species that comprise ecological communities (Danczak et al., 2020).

The most fundamental ecological data type is the species-by-site matrix. This matrix indicates how many individuals of each species occur in each sampled community. Ecologists use species-by-site matrices to ask myriad questions related to biological diversity. Two common analyses are known as α-diversity and β-diversity, and there are numerous metrics for each (Whittaker, 1972; Anderson et al., 2011). α-diversity measures the diversity within a given community. β-diversity has been variously defined, but essentially measures variation in composition across communities. Both α-diversity and β-diversity can be quantified using presence-absence data or they can include estimates of each species' relative abundance within and between communities (Fig. 1).

The chemistry of OM is commonly studied using high resolution mass spectrometry (HRMS) techniques (e.g., Hawkes and Kew, 2020b). Specifically, Fourier transform mass spectrometry (FTMS) techniques are predominantly used, i.e., Orbitrap or Ion Cyclotron Resonance (ICR) MS. At present, the highest resolution approach for untargeted analysis of OM is via a 21 Tesla FTICR MS (Marshall et al., 1998; Bahureksa et al., 2021). The output data produced is a spectrum containing peaks represented by a signal intensity (Fig. 2 y-axis) and a mass-to-charge ratio ($m/z$) (Fig. 2 x-axis), which is equivalent to the mass for singly charged ions as routinely detected in natural organic matter (NOM) measurements. In turn, regardless of the type of MS instrument used, the MS data inherently lead to an OM peak-by-sample data matrix akin to an ecological species-by-site data matrix. The high resolution data from MS often results in a large matrix, wherein a single sample may contain thousands to tens of thousands of peaks. To take advantage of these rich data, HRMS data have been analyzed using the same α-diversity and β-diversity metrics that are commonly used by ecologists to study biological diversity (e.g., Kellerman et al., 2014). This is exciting as it allows the same conceptual questions and quantitative frameworks to be applied to biological (e.g., microbial communities) and chemical (i.e., OM) components that directly interact with each other within ecosystems (Lucas et al., 2016; Osterholz et al., 2016; Li et al., 2018; Tanentzap et al., 2019; Danczak et al., 2020, 2021).

The use of ecological metrics with MS data is particularly common with FTMS datasets, and contains great potential to continue leveraging concepts from ecology in high-resolution OM analyses. However, when FTMS peak intensity data are used in the estimations of α-diversity, β-diversity, and related ecological analyses (e.g., 'species' abundance distributions), potential problems can arise. At the root of these problems lies the assumption that differences in peak intensity are proportional to differences in concentrations of the associated molecules. Consequently, the biases and uncertainties introduced by relying on this assumption are unclear. In certain situations, however, peak intensity-based ecological analyses of MS data can provide valid information—even when the underlying assumption is invalid—and the extent to which such situations exist is likewise unclear. To help advance the robust use of FTMS datasets that has been emerging in environmental science studies over the last few decades, we review the theoretical reasons why between-peak intensities do not correspond to differences in concentration, provide empirical support for our assertions, use *in silico* studies to quantify the associated impacts on ecological analyses,



provide practical recommendations, and propose a path forward that may eventually enable improved usage of
FTMS peak intensities for quantification.
**2 Theoretical Limitations**
To address why peak intensities in FTMS cannot be used to infer between-peak changes in concentration, we review
critical theoretical concepts about mass spectrometry. We focus on FTMS (i.e., FTICR and Orbitrap), but many of
the principles and limitations—especially ionization and ion transmission—are applicable across all MS platforms.
In this section, we highlight three main mass spectrometry considerations: ionization, ion transfer, and ion signal
detection in the context of a generalized commercial FTICR mass spectrometer. Theoretical limitations have two
main practical implications tied to within-peak and between-peak comparisons (Fig. 2). Here, we define within-peak
comparison as comparing the same feature (*m/z* or molecular formula) across different sample spectra (i.e., within
two or more sample spectra), whereas between-peak comparison occurs between different features (*m/z* or molecular
formulas) across the same spectra.
The first implication is that if instrument parameters are kept consistent, within-peak/between-sample biases are
minimized, though between-peak/within-sample biases are inherently unavoidable. The second implication is that
because of inherent sample and matrix variation and subsequent effects, between-peak/between-sample biases can
be significant and may be indeterminable.
**2.1 Ionization Biases**
Electrospray ionization (ESI), the most commonly used technique for generating ions from NOM samples, is a 'soft'
ionization technique that predominantly yields intact molecular ions. Generally, ESI of NOM samples produces
protonated or deprotonated ion types (positive or negative polarity, respectively), which can only be formed if some
pre-existing basic or acidic functionality is available in the molecule to support this. ESI also commonly produces
adduct ions, such as sodium adducts $[M+Na]^+$ in positive mode and chloride adducts $[M+Cl]^-$ in negative mode. The
ionization efficiency of any given molecule depends on its structure, pKa, sample matrix and composition (Kruve et
al., 2014). Ionization suppression occurs when multiple species are present in a sample, and the ionization efficiency
of one analyte is altered by the presence of another (Ruddy et al., 2018). These issues all confound when dealing
with complex samples with unknown compositions. While users can apply controls to account for some matrix
differences (concentration, solvent, pH), the unknown (and unknowable) differences in molecular composition of
complex mixtures cannot be accounted for, and therefore comparison of peak intensities in different samples is
prone to uncertainty.
Importantly, in these highly complex samples, one detected peak commonly combines signals from multiple
different isomers, i.e., the same molecular formula but with a different structure. While the different structural
features impact the ionization efficiency of a given molecule, the recorded spectrum shows the superposition of
these isomers. To date, no liquid chromatography (Kim et al., 2019; Han et al., 2021) or ion mobility separation
(Tose et al., 2018; Leyva et al., 2020) technique has demonstrated sufficient resolution for the most complex of
samples (such as NOM), instead yielding broad distributions in these orthogonal dimensions. Therefore, not only
must we apply extreme caution in inferring chemical properties from molecular formula alone, but we must also be
aware of underlying subtleties that may distort comparisons of molecular formula and peak intensities within
samples such as the presence of isomers.
**2.2 Ion transmission**
Before ions can be detected in the (ion) trap, they must be transmitted from the instrument source to the trap. Ion
transmission, including ion accumulation, is not unbiased. Ions are manipulated through the instrument ion optics,
across differential pressure regimes, using radiofrequency (RF) and direct current (DC) potentials to guide, focus,





and accumulate the ions. The specific values of these parameters have effects on the mass ranges transmitted. Further, the specific timings, geometries, and vacuum regimes all have effects upon ion transmission efficiency and biases. For this reason, quantitative comparison of intensities across widely differing $m/z$ is not directly possible.

In FTMS, packets of ions are accumulated and 'cooled' in a trap prior to their transmission to the analyzer cell (Fig. 3 Panel A section d; Senko et al., 1997; Makarov et al., 2006). The duration of time in which ions are accumulated is varied to yield an optimal ion population for the analyzer cell, which has a finite charge capacity. The duration of this event has been directly observed to change the relative ion populations (Cao et al., 2016). Thus, when balancing the need for controlled ion populations - critical for a high resolution, high fidelity measurement - and minimal variation in ion accumulation time, there is a risk of further biasing the relative ion intensities.

Finally, time-of-flight biases come into play in FTICR MS. Ions are transmitted from the ion accumulation trap to the ICR cell along one or more transfer multipoles (Fig. 3 Panel A section e). The distance between ion accumulation trap and ICR cell may be quite long, e.g., 2.4 meters on a 21 Tesla instrument (Shaw et al., 2016), and therefore the time required for ions to travel this distance (the 'time-of-flight', a millisecond or longer) may cause dispersion in the ion packet (Fig. 3 Panel B). While the packet of ions may leave the accumulation trap simultaneously, because smaller ions travel faster the packet arrives at the analyzer cell as a dispersed distribution of ions. Therefore, only a subset of this population, with regards $m/z$-range and ion energies, is optimally trapped in the ICR cell. Thus, these biases in ion transmission do not allow for quantitative comparison of peak intensities between ions with differing $m/z$ ratios.

**2.3 Ion signal detection**

To directly use FTMS peak intensities quantitatively, we must first understand how those intensities arise and the biases which can affect them. In ion trapping measurements, such as FTICR and Orbitrap MS, the motion of ions within a static magnetic (ICR) or electric (Orbitrap) field induces an image current upon detection electrodes. The frequency of this motion is proportional to the mass-to-charge ratio ($m/z$) of the ion, while the intensity of the signal is proportional to the abundance of the ion in the trap, the proximity of the ion to the electrode (Kaiser et al., 2013), and the charge state of the ion (Wörner et al., 2020). Thus, at a first approximation, the signal intensity between different $m/z$ ions could be compared provided they are excited to the same radius (ICR) and have the same charge state (Fig. 3 Panels B, C). However, these provisions are not always met. Ions with very close frequencies, which are not fully resolved, may affect each other's signal intensity, and the Fourier transform does not allow for extremely accurate relative quantification of ion abundance between peaks (Makarov et al., 2019). With FTICR, most commercial (e.g., Bruker) instruments use a CHIRP, or frequency-swept, excitation pulse which does not excite all ions to exactly the same radii (Kaiser et al., 2013). In addition, while most ions in NOM mass spectra are singly charged, some mass spectra contain multiply charged interferences (Smith et al., 2018; Patriarca and Hawkes, 2021). Still, in both instrument types, signal intensities may be used to describe the ion populations quantitatively provided that the charge states are the same, a flat-excitation profile is used (or the ions are sufficiently close in frequency space such that they are excited to the same radii), and the user clearly understands that the ion population in the trap may not accurately reflect the molecular composition of the sample.

Within a well-designed experiment and a constrained sample set, many of these points may be mitigated. However, objectively proving the degree of mitigation is non-trivial, and there remains great uncertainty about the relationship between peak intensity and molecular concentrations, particularly for complex matrices such as NOM. Furthermore, as shown in a recent interlaboratory study (Hawkes et al., 2020), measuring the same samples with different instrumentation can lead to differing results, thus further highlighting potential pitfalls in quantitative analysis of these data.




## 3 Empirical Limitations

Despite the aforementioned fundamental and theoretical limitations and uncertainties in using peak intensity data, it is still helpful to demonstrate these limitations with real-world empirical measurements. In this section, we demonstrate, with ideal and non-ideal samples, the non-quantitative nature of these measurements.

### 3.1 Direct comparison of signal intensities in idealized samples is problematic

In the ideal case, samples are analyzed with identical matrices, equivalent concentrations for each compound, and free from competitive ionization/ionization suppression (Ruddy et al., 2018). However, even in this ideal case, different molecules ionize with different efficiencies, and thus their signal intensities are not equal. To demonstrate the non-equal response for various analytes in various conditions, we acquired a series of contemporaneous mass spectra of several compounds in different conditions. First, in Fig 4A, we prepared three dilution ladders of three pure compounds - analyzed separately - in pure methanol. Clearly, these three molecules yield starkly different signal intensities for otherwise identical conditions, and thus directly comparing their intensities would not be a valid means to infer their relative concentrations in solution. At an extreme, trehalose, a carbohydrate, yields nearly as little signal at 500ppb as sinapic acid does at 200ppb. Even between the two structures containing a carboxylic acid moiety - a typical indicator of good negative mode ESI response - there is a significant difference in signal intensity. Thus, directly comparing the signal intensities of different ions - even in idealized situations - cannot be used as a proxy for concentration or abundance determination absent a calibration curve.

Subsequently, we highlight the challenge of comparing ions of the same exact mass. Here, in Fig 4B, we again prepared dilution ladders of three pure compounds in methanol, however these are all structural isomers with the same molecular formula and thus exact mass. Again, a stark difference in signal intensity is observed, even between nominally similar structures. This issue is particularly troubling for direct infusion measurements of complex mixtures, where we do not, and cannot, know the structural identity of individual peaks, and instead are limited to molecular formulas. Thus, if we compare peaks with the same exact mass, same molecular formula, between different samples, we cannot be sure that they are the same molecule, and thus again comparing their signal intensities as a proxy for abundance is problematic. Additionally, structural isomers can have vastly different ecological/biogeochemical function, and therefore this consideration is important to note for subsequent interpretations of NOM samples. Further complicating this issue is the known fact that in highly complex mixtures like organic matter, most - or all - peaks are actually the superposition of multiple different isomeric compounds. Demonstrated by chromatography (Kim et al., 2019) or ion mobility separations (Leyva et al., 2019), or by statistical inference of tandem mass spectrometry (Zark et al., 2017), each peak may be several isomers of various relative intensities. Thus, even if the same isomers were present across samples, it cannot be known that their relative abundances are the same - and again, it is problematic to directly compare the intensities of signal corresponding to nominally the same molecular formula across different mass spectra.

One caveat with the above experiments, of course, is that it is a direct infusion measurement. The chemicals used were nominally pure, but any trace impurity - either from their production and isolation, or from sample preparation - may impact the measured signal intensity. Which leads us to the next point - matrix effects are intrinsically challenging to control for, and have significant impacts on mass spectra.

### 3.2 Matrix effects substantially impact signal intensities in complex mixtures

Of course, analyses are often performed on complex mixtures, containing a diverse range of thousands of molecules of unknown structures and relative concentrations. Furthermore, samples often contain 'inorganic' interferences, such as salts. Routinely, scientists will desalt samples with solid phase extraction, but such processes can leach impurities into the sample, don't necessarily remove all interferences, and can remove select pools of NOM due to their functionalities, depending on the sorbent or resin (Raeke et al., 2016; Li et al., 2017). As such, real world non-





ideal samples contain a multitude of matrix effects and sources for ionization suppression, or adduct formation,
which yield spectra that are even more challenging to quantitatively interpret.

To explore the impacts of matrix effects (Fig. 4C-E), we prepared solutions of six different pure compounds at a
fixed concentration (100ppb) in three different solvent systems - pure methanol, methanol from elution off of a
BondElut SPE cartridge, and methanol from elution off of a BondElut SPE cartridge which had been loaded with
artificial river water (ARW). Additionally, we added a complex mixture - Suwannee River Fulvic Acid (SRFA), at
six different concentrations, to each sample. Again, samples were analyzed independently but contemporaneously
on the same instrument to mirror a real study.

In methanol only solvent, with no addition of SRFA, the six compounds - as expected - yield different signal
intensities (Fig. 4C), further confirming what was previously observed. As the concentration of SRFA is increased to
2 ppm, the relative signal intensity increases for some of these analytes - possibly as a function of endogenous
molecules with the same molecular formula as those spiked in - but decreases for others. Above 2 ppm of SRFA,
however, all signals for our reference compounds are substantially decreased, most likely as a result of competitive
ionization effects of the addition of the complex mixture.

Use of an 'impure' methanol solvent, i.e., the eluent from a SPE blank (Fig. 4D) or from an SPE of artificial river
water (Fig. 4E), results in even more ionization suppression and differential signal response. In both cases, the
maximum signal intensity is only 20% of what was seen in pure methanol (Fig. 4C), indicating that the leachate or
residual salts from the SPE protocol impacted sensitivity. Further, here only two analytes (aesculin and chlorogenic
acid) ionize well at all, with the other 4 yielding poor or no signal. Addition of SRFA, again, decreases signal
intensity, though at 40 ppm SRFA some minor features increase, likely due to endogenous features with the same
molecular formula as our standards.

Cumulatively, empirical evidence and instrumental theory demonstrate that it is not possible - with direct infusion
measurements of complex mixtures - to directly compare signal intensities as a proxy for molecular abundance
between different peaks within a spectrum, or between the same peak across spectra, even in idealized cases.
Strategies to use calibration curves will fail due to unknown structural composition, and established normalization
techniques cannot factor in the large range of sources of experimental variation. That said, there may be cases where
a high-level comparison of trends can yield valid semi-quantitative comparisons between spectra, relying on a
statistical aggregation of individually unreliable trends. Additionally, modeling of constrained systems may allow
for improved, data-driven and mechanistic based machine-learning data normalization strategies.

**4 Conceptual implications for use of ecological metrics**

The preceding sections have shown both theoretically and empirically that there are challenges to using HRMS peak
intensities as proxies for relative changes in concentrations of organic molecules. The implication is that there may
be specific kinds of ecologically-inspired analyses (e.g., Fig. 1) that are or are not appropriate to use with HRMS
peak intensity data. To understand what may or may not be a valid analysis, it is critical to differentiate analyses into
two classes: those based on within-peak intensity comparisons and those based on between-peak intensity
comparisons (Fig. 2). As noted above, within-peak is based on comparing the same feature (*m/z* or molecular
formula) across spectra/samples, whereas between-peak compares different features (*m/z* or molecular formulas)
across and within spectra/samples.

Analyses using between-peak intensity comparisons are the most likely to be problematic. To help clarify why this
is, consider an ecological setting in which a researcher aims to quantify α-diversity and β-diversity (Fig. 1) of tree
communities (Fig. 5, left-side). The researcher will likely set up a plot of a given size and then directly count the



number of each tree species in each plot. This generates the species-by-site matrix filled with directly observed
abundance counts for each species. In such a situation, the ability of the researcher to observe individuals of each
species does not vary appreciably across species because each tree is not moving and our ability to see it is not
influenced by environmental factors. In turn, the number of individuals observed for a given species is quantitatively
comparable to the number of individuals observed for all other species in the plot. The assumption that differences
in observed abundances carry robust information about differences in actual abundances is thus supported. In turn, it
is valid to use relative abundances to compute α-diversity such as via Shannon evenness (Elliott et al., 1997;
Mouillot and Leprêtre, 1999; Redowan, 2015). Furthermore, the ability to observe each species is the same across
communities. In turn, it is valid to use relative abundances to compute β-diversity (e.g., via Bray-Curtis; Anderson et
al., 2011) or conduct any other ecological analysis that uses abundance data (e.g., species abundance distributions
McGill et al., 2007).
We contrast this tree community example with another ecological setting. Consider a researcher studying bird
communities (Fig. 5, right side) that estimated species abundances solely based on the number of times an observer
hears the call of a given species. In this case, those species that call more frequently and/or more loudly (more likely
to be heard), will be inferred to have higher abundance even if all species in the community have the same
abundance. That is, such a method generates data that may indicate which species are present, but the 'call counts'
do not carry reliable information regarding absolute or between-species relative abundances. Follow-on analyses of
α-diversity and β-diversity should, therefore, be limited to approaches that use presence/absence data, and species
abundance distributions cannot be quantified.
If we continue with the bird community example and assume that the detectability of a given bird species is
consistent across sampled locations (or times), then it would be appropriate to examine variation in within-species
call counts. This within-species analysis is directly analogous to the HRMS within-peak time series analysis in
Merder et al. (2021), discussed below. However, if call counts of a given species are suppressed by the
presence/abundance of other species, then call counts of a given species do not indicate an increase in its abundance.
This is directly analogous to influences of the OM matrix: if the presence/abundance of a given organic molecule
modifies the ionization of other molecules, then within-peak changes in intensity do not indicate changes in their
concentrations. In turn, analyses based on within-peak intensity comparisons are not always valid, especially if there
are significant cross-sample changes in the OM matrix.
Unfortunately, as demonstrated in the previous sections, HRMS data align with the bird community examples and
never reflect the tree community example. The unique chemistry of every molecule fundamentally results in
different ionization properties for other molecules. Thus, the differing physics of each molecule strongly influences
between-peak differences in peak intensity. Those molecules that ionize more easily result in higher peak intensities,
which is akin to bird species that call more frequently or more loudly resulting in a larger number of 'call counts.' In
turn, between-peak differences in intensity cannot be used as a proxy to indicate between-peak differences in
concentration. This could invalidate the application of ecological metrics that use between-peak differences in
intensity.
In contrast to between-peak comparisons, within-peak comparisons examine changes in relative intensity of a single
peak across samples. Such within-peak comparisons may be repeated independently for each peak of interest in a
given dataset. For example, Merder et al. (2021) quantified temporal dynamics of individual HRMS peaks and then
binned peaks into different groups with characteristic temporal fluctuations. In those analyses, peak intensities were
not compared between peaks. Instead, the temporal dynamics of each peak was compared to temporal dynamics of
other peaks. The underlying assumption of this type of analysis is that a between-sample increase in the intensity of
a given peak can be used as a robust proxy of a between-sample increase in concentration of that peak. Materials
presented in the previous sections indicate that this assumption can be met in some instances when using HRMS





data. However, great care is required with strong attention paid to assumptions of analysis methods. For example,
using Pearson correlation makes the assumption that concentration of a given peak is a *linear* function of changes in
its peak intensity. We showed above (Fig. 4) that assumption is not always valid even in ideal conditions. Using a
Spearman correlation avoids this assumption because it is based on ranks. That is, using Spearman (e.g., Kellerman
et al., 2014) makes the more realistic assumption (for FTICR MS data) that an increase in concentration of a given
peak is reflected as an increase in its peak intensity, without assuming any statistical or mathematical form of that
relationship.
**5 Quantitative impacts**
The previous sections show that between-peak changes in peak intensity do not accurately reflect between-peak
changes in abundance (Fig. 4). This violates a fundamental assumption of abundance-based ecological analyses:
proxies of abundance (e.g., peak intensity) must reflect actual abundance. In turn, it is tempting to infer that mass
spectrometry peak intensities cannot be used at all in ecological analyses. However, the impacts of violating the
assumption have not been directly quantified. This is a significant gap considering the growing number of
publications that use peak intensities to compute abundance-based ecological metrics over the last couple decades.
Therefore, there is a need to quantitatively understand biases and uncertainties introduced in ecological metrics
(e.g., α and β diversities) and/or models when peak intensity does not reflect abundance or concentration. To
provide an initial evaluation, we developed an *in silico* simulation model that generates synthetic data, introduces
specific kinds of error commonly found with HRMS datasets (discussed above in detail), and computes within-
sample (e.g., Shannon diversity) and between-sample (e.g., Bray-Curtis) ecological metrics (Fig. 6). This allows for
comparison between true values of the metrics and the values observed after each type of error is introduced, which
is impossible to do with non-simulated datasets. To generate synthetic data, we randomly assigned abundances to
either 100 or 1000 peaks. Abundances were sampled with replacement from a Gaussian distribution that varied in
mean and standard deviation across synthetic samples and across simulation iterations. Abundances were drawn
twice to generate two independent samples per simulation, and the simulation was run 100 times for each number-
of-peaks (100 or 1000 peaks per sample; referred to below as 'peak richness'). The reason for variation in the
Gaussian distributions was to generate synthetic samples that varied in composition within and across simulations to
ensure that the ecological metrics (see below) would vary across simulations. This was necessary to evaluate how
biases in the metrics varied across a broad range of metric values.
We simulated two types of error, and both can be representative of variation in ionization efficiency. The goal was
to generate synthetic data that mimicked our empirical and theoretical observations in the sense that observed peak
intensities did not reflect true abundances. For each type of error and within each iteration of the simulation, the
error was introduced 100 times (i.e., 100 error iterations were nested within each sample-generation iteration). The
first type of error was designed to diminish the between-peak relationship between observed intensity and true
abundance. For this we multiplied the true abundance of each peak by a random number drawn from a uniform
distribution ranging from 0 to 100. For each peak we multiplied the same random error to its abundance in each of
the two synthetic samples within each iteration. The error-modified abundance of each peak in each synthetic
sample was considered to be the observed peak intensity.
As expected, introducing error resulted in a relatively weak relationship between observed intensity and true
abundance, with the amount of error increasing with true abundance (Fig. S1) and a median $R^2$ of ~0.5 (see black
line in Figure 7). Between-peak differences in observed intensity were also weakly related to between-peak
differences in true abundance (Fig. 8A), with a median $R^2$ of ~0.5 (see blue line in Figure 7). Because the same
peak-level error-factor was used across both synthetic samples within a given simulation iteration, the within-peak





between-sample differences in observed intensity were relatively strongly correlated to within-peak between-sample
differences in true abundance (Fig. 8B), with a median $R^2$ of ~0.75 (see the gray line in Figure 7).
The second type of error introduced represents situations in which there is variation in ionization efficiency across
molecules – as in the first type of error – but that ionization efficiency also varies across samples. Molecules may
vary in their ionization efficiency across samples due to changes in the composition of organic molecules and/or
changes in inorganic solutes. In this case, ionization efficiency of any given molecule is due to interactions with
other organic and inorganic molecules within a given sample. For this, we multiplied the true abundance of each
peak by a random number drawn from a uniform distribution ranging from 0 to 100. For each iteration of the
simulation this was done independently for both synthetic samples. In this way, ionization efficiency for a given
peak in a given synthetic sample was independent of its ionization efficiency in the other synthetic sample. In turn,
the error-modified abundance of each peak in each synthetic sample was considered to be the observed peak
intensity.
We observed a relatively large influence of allowing ionization efficiency to vary randomly across samples. That is,
the within-peak between-sample differences in observed intensity were relatively weakly correlated to within-peak
between-sample differences in true abundance (Fig. 8B), with a median R2 of ~0.5 (see the red line in Figure 7).
Comparing this to the same relationship that emerged under the first type of error shows a much weaker relationship
when ionization efficiency varies between samples (compare the gray and red lines in Figure 7). This is expected as
variation in ionization efficiency will add random noise to the within-peak between-sample differences in observed
peak intensity. We note that variation in ionization efficiency is independent between peaks for both the first and
second types of error. The between-peak relationship summarized in Figure 7 (blue line) is, therefore, equivalent for
both types of error, which is also shown by the strong similarity between Figure 8A and 8C.
To examine influences of both types of error on ecological metrics we used the initial true abundances and the error-
modified abundances (i.e., observed intensity values) to calculate true and 'observed' values of within-sample
Shannon diversity and between-sample Bray-Curtis. We also assigned a trait value to each peak and calculated true
and observed sample-level mean trait values; the mean values for each sample were weighted by true abundance
(true mean) or observed intensity (observed mean). To evaluate biases and uncertainty introduced by both types of
error we regressed observed values for each metric against their true values. This was done independently for each
level of peak richness to evaluate how bias and uncertainty vary with the number of peaks contained within a
sample.
Relating 'observed' values of each metric to their true values revealed that the patterns observed in peak-intensity-
based ecological metrics are likely to be qualitatively robust, even though quantitative biases do exist (Figs. 9-11).
All three ecological metrics showed monotonic relationships between observed and true values. Uncertainty was
lower when samples had 1000 peaks, relative to samples with 100 peaks; in Figures 9-11 all A/B and C/D panels
have 100 and 1000 peaks, respectively. Monotonic relationships and lower uncertainty with more peaks were found
for both within-sample and between-sample error; in Figures 9-11 all A/C and B/D panels have within-sample and
between-sample errors, respectively. For Shannon diversity, observed values were consistently lower than true
values, but all observed vs. true relationships were linear (Fig. 9). For Bray-Curtis, inclusion of between-sample
error resulted in an overestimation of values and non-linear (but monotonic) relationships between observed and true
values (Fig. 10). For mean trait values, the observed values had uncertainty but there were no systematic quantitative
biases and the relationships between observed and true values were consistently linear (Fig. 11). Furthermore, the
variation in observed values explained by true values (via a linear model) increases rapidly with the number of
peaks, and sharply asymptotes beyond ~500-1000 peaks per sample (**Fig. S2**). We caution that the number of peaks
needed to reach the asymptote, thereby minimizing error, is likely dataset dependent and 500-1000 peaks should not
be taken as a general rule.



## 6 Conclusions and Recommendations

There is increasing interest in using ecological metrics to study organic matter chemistry across a broad range of environments and settings. It is vital that this growing body of work be based on rigorous use of the data to develop trust in the associated conceptual and mechanistic inferences. To do so requires deep understanding of the metrics themselves, full awareness of the limitations of the OM data from mass spectrometers, and careful use of the metrics informed by the data limitations. We suggest that studies/publications that use peak intensities need to include material that directly discusses the data limitations, what peak intensities do and do not represent (e.g., tree-like vs. bird-like data; Fig. 5), and how knowledge of those limitations was used to select specific metrics.

We have provided both strong theoretical reasoning and empirical observations showing that peak intensities do not directly map to concentrations of the associated organic molecules within complex mixtures of organic molecules. This is particularly true for between-peak comparisons of intensity, and statistical post-hoc normalizations of peak intensity data do not solve this problem. That is, there are no situations that we are aware of in which between-peak differences in intensity indicate between-peak differences in concentration. We therefore assert that between-peak differences in intensity within HRMS data cannot be used to make direct inferences related to between-peak variation in abundance or concentration. This means that HRMS data are unlikely to provide informative ecological analyses based directly on variation in species abundances. In particular, estimation of 'species abundance distributions' is likely to be invalid. Analyses that bin peaks into high and low abundance groups based on between-peak differences in concentration are, likewise, almost certainly invalid. We did not directly evaluate these types of analyses, however, and we suggest that future work should expand upon the ecological metrics examined here via simulation.

While certain ecological analyses of HRMS data are likely to be invalid, we found good performance of some common metrics. These metrics were originally designed to use relative abundances of biological species. Our simulation modeling indicated that at least some α-diversity, β-diversity, and functional trait metrics are likely to provide valid qualitative patterns. That is, conceptual and mechanistic inferences are likely to be valid when based on analyses such as comparing peak-intensity-based ecological metrics across experimental treatments or variation along environmental gradients. The performance of intensity-weighted mean trait values was particularly good in terms of both qualitative and quantitative aspects. We emphasize that we studied a small set of metrics (Shannon diversity, Bray-Curtis, and intensity-weighted trait values) and our inferences only extend to these metrics. Fortunately, it is relatively straightforward to extend the simulation model to additional metrics (e.g., Hill numbers; Hill, 1973) and analyses (e.g., species abundance distributions; McGill et al., 2007) and we suggest that users of such datasets wanting to use additional ecological metrics/analyses test them using simulation models before applying to real-world datasets to ascertain if these metrics hold given the known biases in these analyses and metrics.

To enable robust use of HRMS peak intensity data in future studies, we recommend use of and further development of the simulation model developed here. The simulation model is the only tool we are aware of that can provide objective guidance on what analyses are not valid and the level of uncertainty associated with valid analyses. It should not be taken as a static or mature tool, however. The model should be expanded in a number of ways by including additional ecological metrics/analyses, more than two-sample situations, other ways of modeling error, and measured levels of error between concentrations and peak intensities. This will allow each study to customize the model for their specific application. It should be possible to include the number of samples, the number of peaks in each sample, the peak intensity distributions, number of replicates, and the specific ecological analyses that will be applied. In turn, simulation model outcomes can provide objective guidance tailored to each study. One may think of the resulting guidance as akin to a power analysis whereby the simulation can indicate what can and cannot be inferred from a given dataset. For example, the model indicates that observed Bray-Curtis values have little to no correspondence to true values when Bray-Curtis is below ~0.2 (Fig. 10B, D). Bray-Curtis near and below ~0.2 are



commonly observed in HRMS studies (e.g., Hawkes et al., 2016; Derrien et al., 2018; Bao et al., 2018), and this
disconnect between observations and truth is maintained even with 1000 peaks per sample (Fig. 10D). In turn,
HRMS studies that observe Bray-Curtis below ~0.2 may not be able to use those observations to make valid
conceptual inferences. However, quantitative guidance must be developed for each study and we recommend that a
version of the simulation model should be used by all future studies using peak intensities to conduct ecological
analyses of HRMS data. It may be that in time we understand the general rules well enough to leave the simulation
behind, but for now, failing to use it (or a similar tool) leaves analyses open to criticism and potentially spurious
inferences.
In addition to further use and development of the simulation model, we recommend translation of other modeling
approaches for use with HRMS data. Two potential approaches are based in machine learning and hierarchical
modeling. Machine learning could be used within very tightly controlled systems to understand the magnitude and
nature of non-quantitative biases that disconnect peak intensity from concentration. In this case, one could model the
instrument response for a diverse chemical space in typical environmental samples to learn how measured signal
intensities may relate to starting concentrations. Even if such a model does not yield high-accuracy results, it may
nonetheless help understand error, biases, and robust use of peak intensity data. Furthermore, such a model would be
constrained to the system it was built around, and application outwith this system could be wrong. Potentially in
concert with machine learning, hierarchical modeling could be translated from its application in ecological analyses
(Iknayan et al., 2014) for use with HRMS. This approach has been used to model sources of error that lead to
variation in detectability across biological species, such as variation in species visibility (e.g., Dorazio and Royle,
2005). In turn, data can essentially be corrected by accounting for the modeled sources of error (Roth et al., 2018),
even revealing 'hidden diversity' (Richter et al., 2021). Machine learning could be used to understand sources of
error and, in turn, inform hierarchical models aimed at improving the mapping between peak intensity and
concentration. If successful, this would increase the quality of information provided by peak intensities in both
existing and future datasets, thereby enabling more robust conceptual and mechanistic inferences.
In summary, HRMS has many strengths and weaknesses just like any analytical platform. Careful use of peak
intensity data informed by objective, model-based guidance can overcome some of its weaknesses. Despite peak
intensities not reflecting concentrations, ecological metrics overall appear to perform well. This is likely due to the
law of large numbers as HRMS, especially FTICR MS, datasets often contain 1000 or more peaks per sample. Our
simulation results indicate that large numbers of identified peaks allow ecological metrics to essentially track
towards their true value. We are encouraged by this outcome and look forward to further applications of ecological
metrics, concepts, and theory to organic matter chemistry.

**7 Code Availability**

R code for running the simulation models is available on GitHub: https://github.com/stegen/Peak_Intensity_Sims.
Python code used to process the empirical data and to generate the associated figures will be available upon
publication.

**8 Data Availability**

Raw and processed data will be made publicly available upon manuscript acceptance.

**9 Author Contributions**

WK contributed to conceptualization, experimental data curation, formal analysis, methodology, software,
visualization, writing-original draft, writing-review/editing; AMP contributed to conceptualization, methodology,



visualization, writing-original draft, writing-review/editing; CHC and SMC contributed to investigation and writing-
review/editing; JE contributed to sample preparation and writing-review/editing; MMT contributed to
conceptualization, methodology, writing-review/editing; JH contributed to conceptualization and writing-
review/editing; RKC contributed to project administration, conceptualization, experimental data curation,
methodology, writing-review/editing; JCS contributed to conceptualization, simulation data curation, formal
analysis, funding acquisition, investigation, methodology, software, visualization, writing-original draft, writing-
review/editing.
**10 Competing interests**
The authors declare that they have no conflict of interest.
**11 Acknowledgements**

A portion of this research was performed on a project award (doi:10.46936/intm.proj.2020.51667/60000248) from
the Environmental Molecular Sciences Laboratory, a DOE Office of Science User Facility sponsored by the
Biological and Environmental Research program under Contract No. DE-AC05-76RL01830. JCS was also
supported by an Early Career Award (grant 74193) to JCS at Pacific Northwest National Laboratory (PNNL), a
multiprogram national laboratory operated by Battelle for the United States Department of Energy under contract
DE-AC05-76RL01830. We thank Alan Roebuck for useful feedback on the manuscript, Nathan Johnson for
graphics development, Charles T. Resch for supplying the artificial river water, Patricia Miller and Jason Toyoda for
lab support.

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



**Figures**

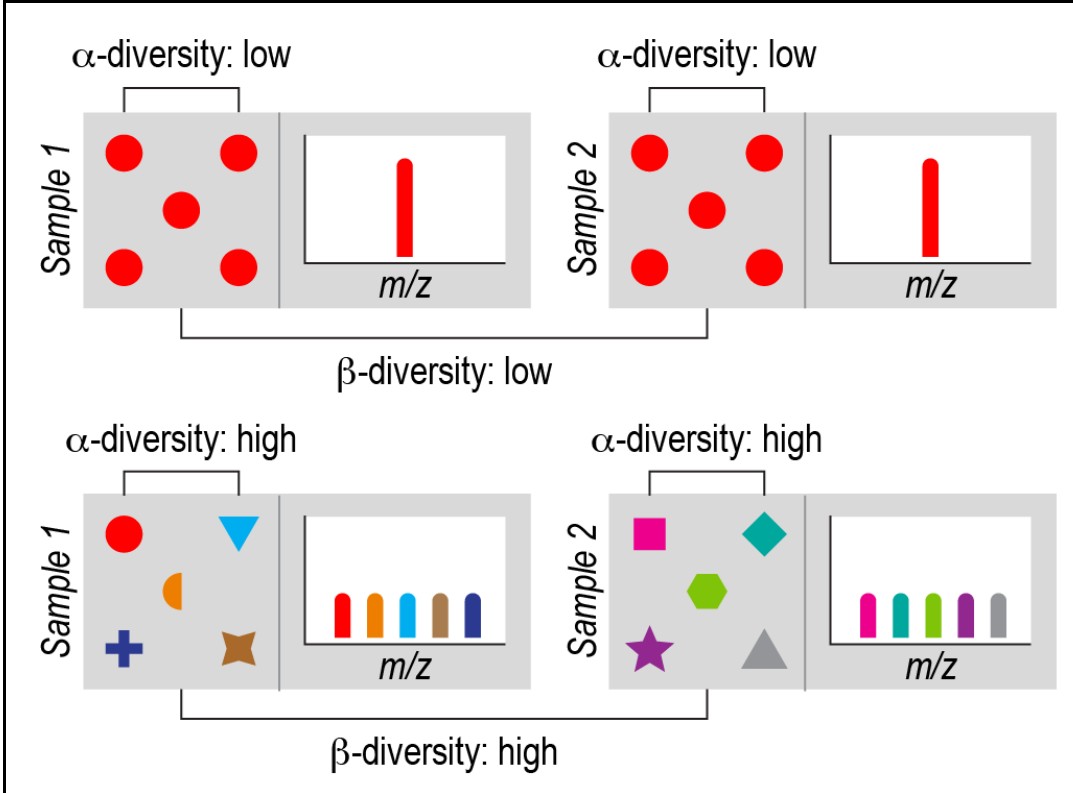

**Figure 1. Ecological concepts of α–diversity and β-diversity.** Each gray box represents a sample of an ecological community or collection of organic molecules (i.e., an OM assemblage). Symbols represent individual organisms or molecules. Different biological or molecular species are represented by a combination of shape and color. (Top) Each sample has one biological species (red circles) or one chemical species (red bar), and the species are the same within and between the samples. This reflects minimal α–diversity because there is a single species. This also reflects minimal β-diversity because there is no difference in which species are present in each sample. (Bottom) Each sample has five species (biological or chemical) represented by different colors and symbols. There are no shared species between samples. This reflects maximum α-diversity because every individual is a different species within each sample, and maximum β-diversity because there are no species shared between samples. In real ecological and OM samples, α–diversity and β-diversity fall between these extremes.






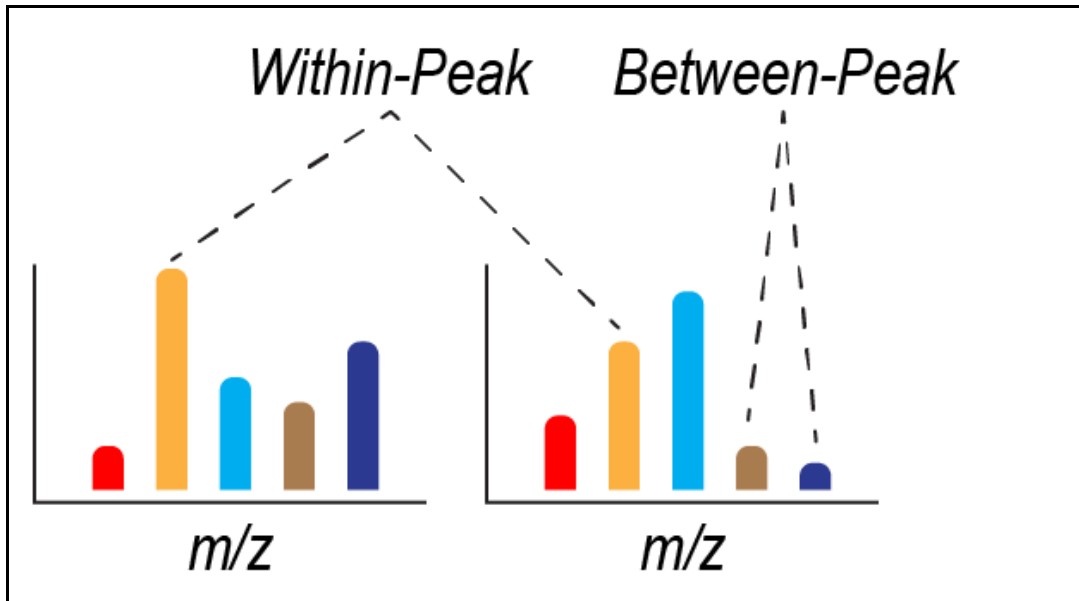

**Figure 2. Summary of within-peak and between-peak comparisons of peak intensity.** Two idealized mass spectra (i.e., from two samples) are shown with each peak defined by a mass-to-charge ratio (*m/z*) and represented by a different color. The intensity of each peak in each sample is represented by the height of each colored bar. Within-peak comparisons of intensity are based on comparing intensities at the same *m/z* across two or more samples. Between-peak comparisons of intensity are based on comparing intensities at two or more *m/z* values. Between-peak comparisons can be done within a sample (as shown) or between samples (not shown).






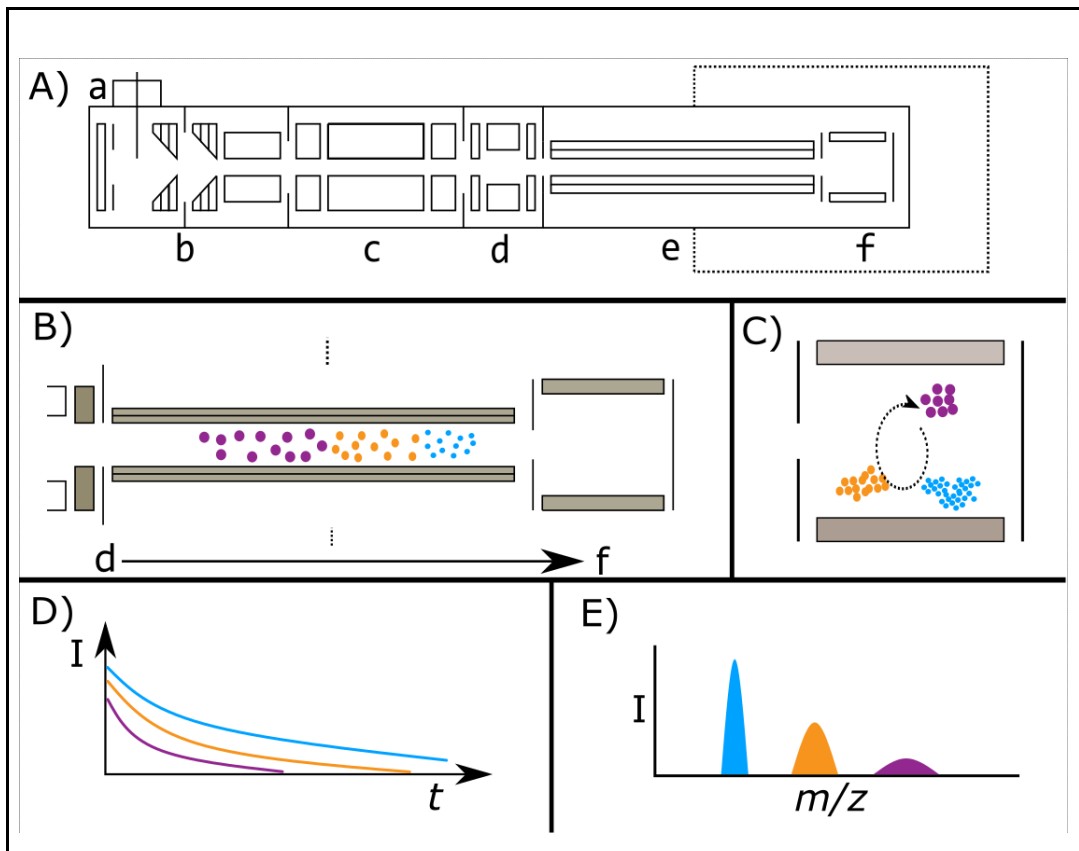

**Figure 3. Illustrative example of a generic FTICR mass spectrometer (panel A), showing common and key biases between FTICR signal intensity and *m/z* of ions (B-E).** Panel A shows the major elements of a generic FTICR mass spectrometer (based loosely on a Bruker solariX FTICR MS geometry). Panel A elements include; a - atmospheric pressure ionization source (i.e. ESI source), b - source ion optics (i.e. dual ion funnels), c - mass selecting quadrupole, d - collision cell, e - transfer multipoles to ICR cell, f - ICR cell. Dashed line indicates the magnetic field. Note: diagram is deliberately simplified and not to scale. Panel B) demonstrates the time-of-flight bias along the transfer multipoles (e) in the 'flight tube', from the collision cell (d) to the ICR cell (f). Lower *m/z* ions travel faster, as indicated by the smaller icons reaching the ICR cell first. Ions are shaded to aid visualization. Panel C) visualizes the effect of a variable excitation radii for ions of different masses, as may happen with a CHIRP excitation pulse. Lower *m/z* ions are closer to the detection electrodes (shaded in gray) and therefore will induce a larger image current. Note also the ion populations have been adjusted from B) to indicate biases from the time-of-flight effect. Panel D) shows the time-domain recorded signal intensity against time, with the ions having an initial intensity roughly proportional to the number of ions in that cloud. However, as time progresses the less abundant ion clouds lose coherence and destabilize more rapidly, resulting in an attenuation of their signal. Note that the real signal would follow a damped sinusoidal function; here an absolute value approximation is shown for simplicity. Panel E) shows the mass spectrum post-Fourier transform, demonstrating that the impact is not only on intensity (peak height), but also resolution (peak width). In all cases, effects are deliberately exaggerated and not-to-scale to aid interpretation.






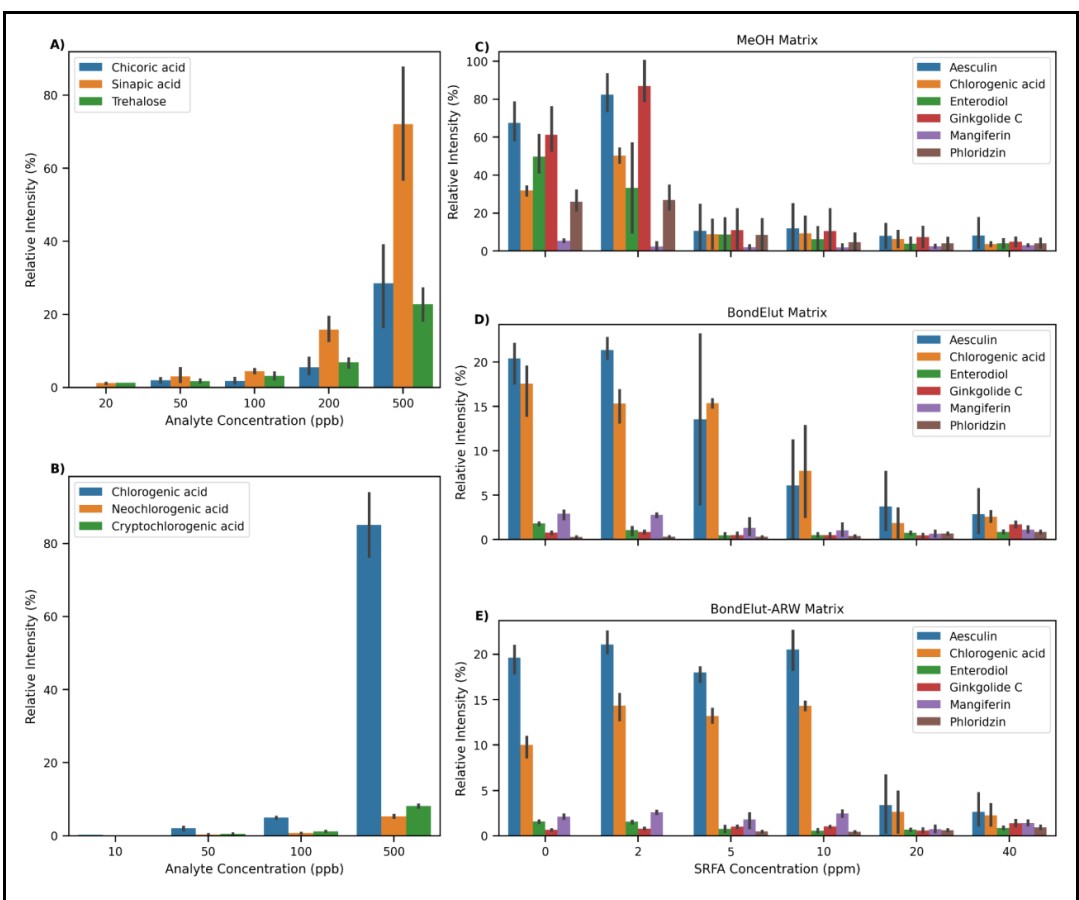

**Figure 4** - A) Barplot visualization of the relationship between signal intensity (relative intensity) and concentration of analyte for three chemically distinct molecules analyzed contemporaneously but independently in pure methanol solvent. B) As with A), but for three structural isomers of chlorogenic acid. C-E) Compounds spiked into three different solvent matrices (methanol, BondElut methanol, and BondElut artificial river water (ARW)) at a fixed concentration (100ppb), but with addition of SRFA at varying concentrations from 0 to 40ppm. In all cases, [M-H]- ion only is shown, but other ions (i.e. [M+Cl]-) were detected. 95% confidence intervals represent the results of triplicate measurements. Intensities have been scaled per plot for A and B, and are on the same scale for C-E).






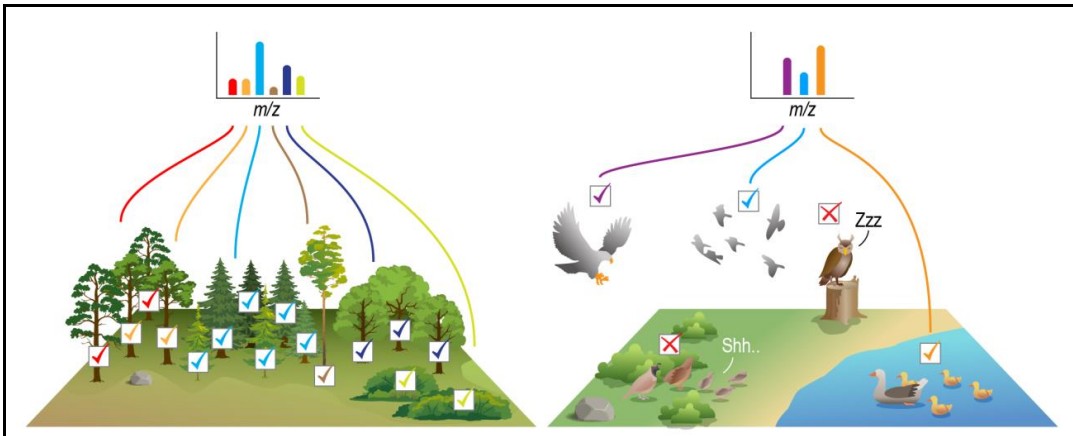

**Figure 5. Graphical summary of how HRMS peak intensity data are often treated (left), which is distinct from the reality of those data (right).** When surveying the number of individuals of each species within a tree community, there is good confidence that the measured abundances are close to real abundances. This is because there is relatively little variation across species in the ability to detect individuals. HRMS peak intensity data are often used as though they are like tree-community data. However, HRMS data are more like bird-community data. That is, the ability to detect different species varies due to intrinsic factors (e.g., activity patterns, how loud and often birds call, etc.) and extrinsic factors (e.g., habitat structural complexity, predator-induced behavioral changes, etc.). Similarly, the intrinsic physics of a given molecule will impact its ability to ionize and thus its observed peak intensity, and in environmental samples there are thousands of molecular species that impact the ionization 'behavior' of each other. HRMS data being more bird-like than tree-like needs to be accounted for when performing ecological analyses using HRMS data.






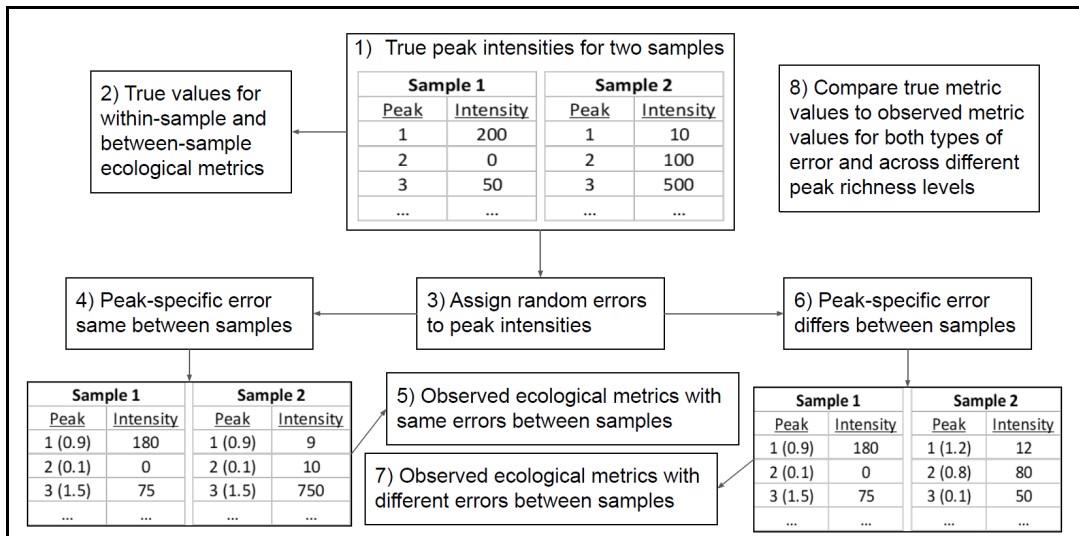

**Figure 6. Flow diagram for the *in silico* simulation model.** The model was used to evaluate how ecological metrics are impacted by variation in ionization across organic molecules (i.e., peaks). The true peak intensities are what is expected if intensity is linearly to concentration, and all peaks fall along the same linear function. Variation in ionization adds error around this idealized linear relationship. The error is modeled in two ways: the error applied to a given peak is either the same between samples (i.e., there are no variable matrix effects on ionization) or varies randomly between samples (i.e., there are variable matrix effects on ionization). In the lower tables the proportional error applied to each peak is provided parenthetically. The tables are for demonstration and show only three peaks per sample. The number of peaks per sample was set to either 100 or 1000.







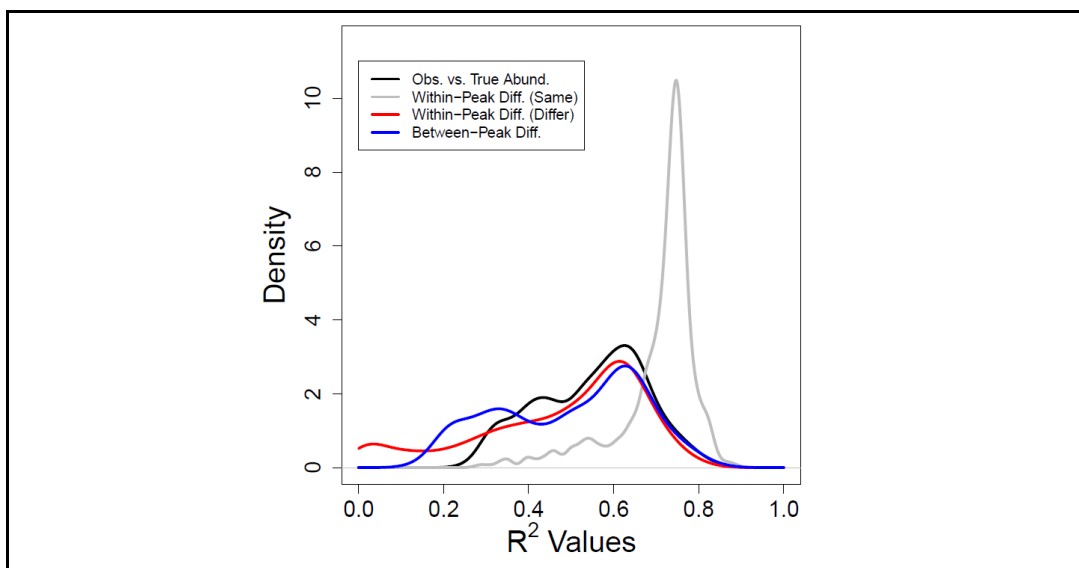

**Figure 7. Variation in observed intensity explained by true abundance.** Kernel density functions are shown for different relationships and types of error. Density functions were fit using $R^2$ values collated from across simulation iterations. Higher $R^2$ values indicate a stronger link (i.e., lower uncertainty) between observed intensities and true abundances. Black is for the relationship shown in Figure S1. Blue is for between-peak within-sample differences (example relationships shown in Figures 8A,C). Gray is for within-peak between-sample differences when the same peak-level error was used for both synthetic samples within a given simulation iteration (example relationship shown in Figure 8B). Red is for within-peak between-sample differences when different peak-level error was used across the synthetic samples within a given simulation iteration (example relationship shown in Figure 8D). While there are central tendencies in all four distributions, there is also significant variation in the degree to which observed intensities reflect true abundances.




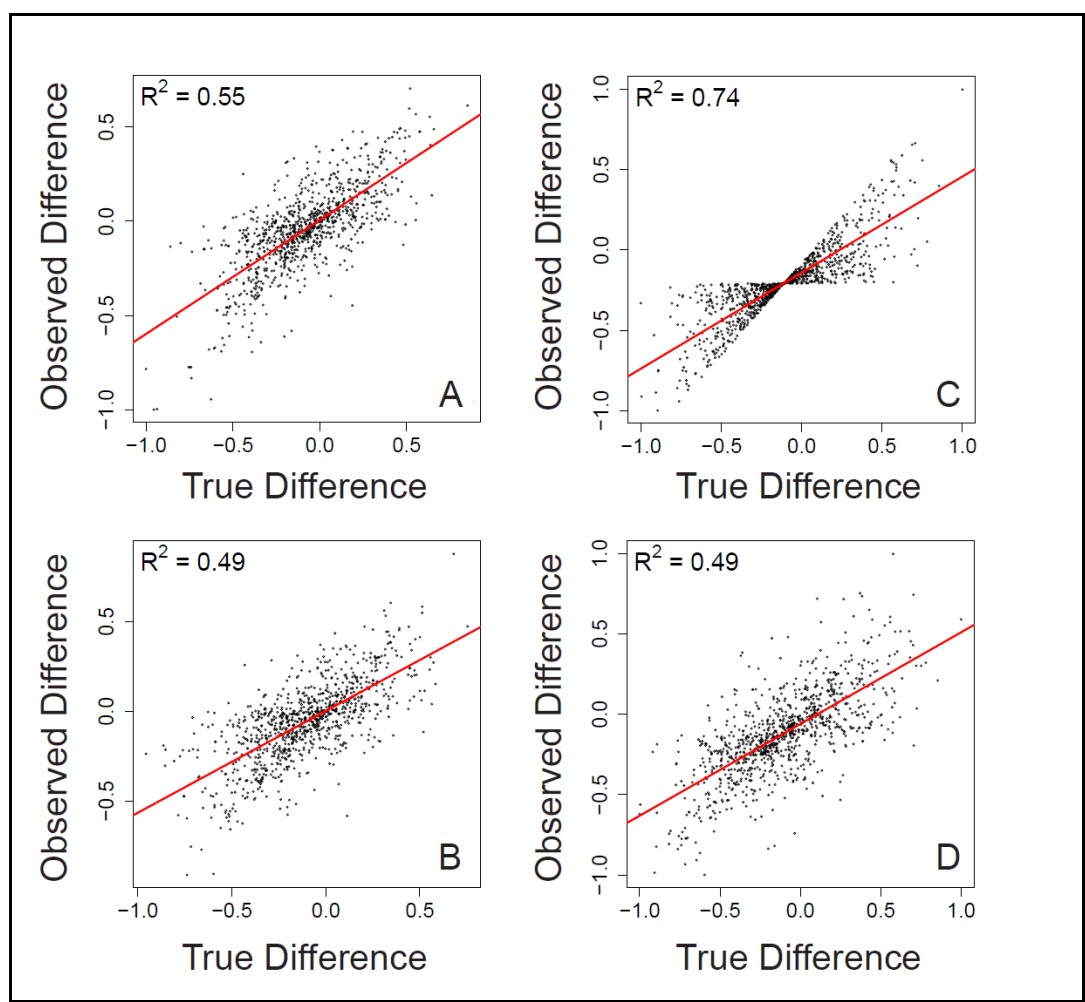

**Figure 8. Observed differences in peak intensity as a function of true differences in peak intensity across both within-peak and between-peak comparisons and across both kinds of error.** (A) Between-peak differences with the same error applied to a given peak between samples. (B) Within-peak differences with the same error applied to a given peak between samples. (C) Between-peak differences with different errors applied to a given peak between samples. (D) Within-peak differences with different errors applied to a given peak between samples. On all panels the red line represents the linear regression model, and the associated $R^2$ value is provided.






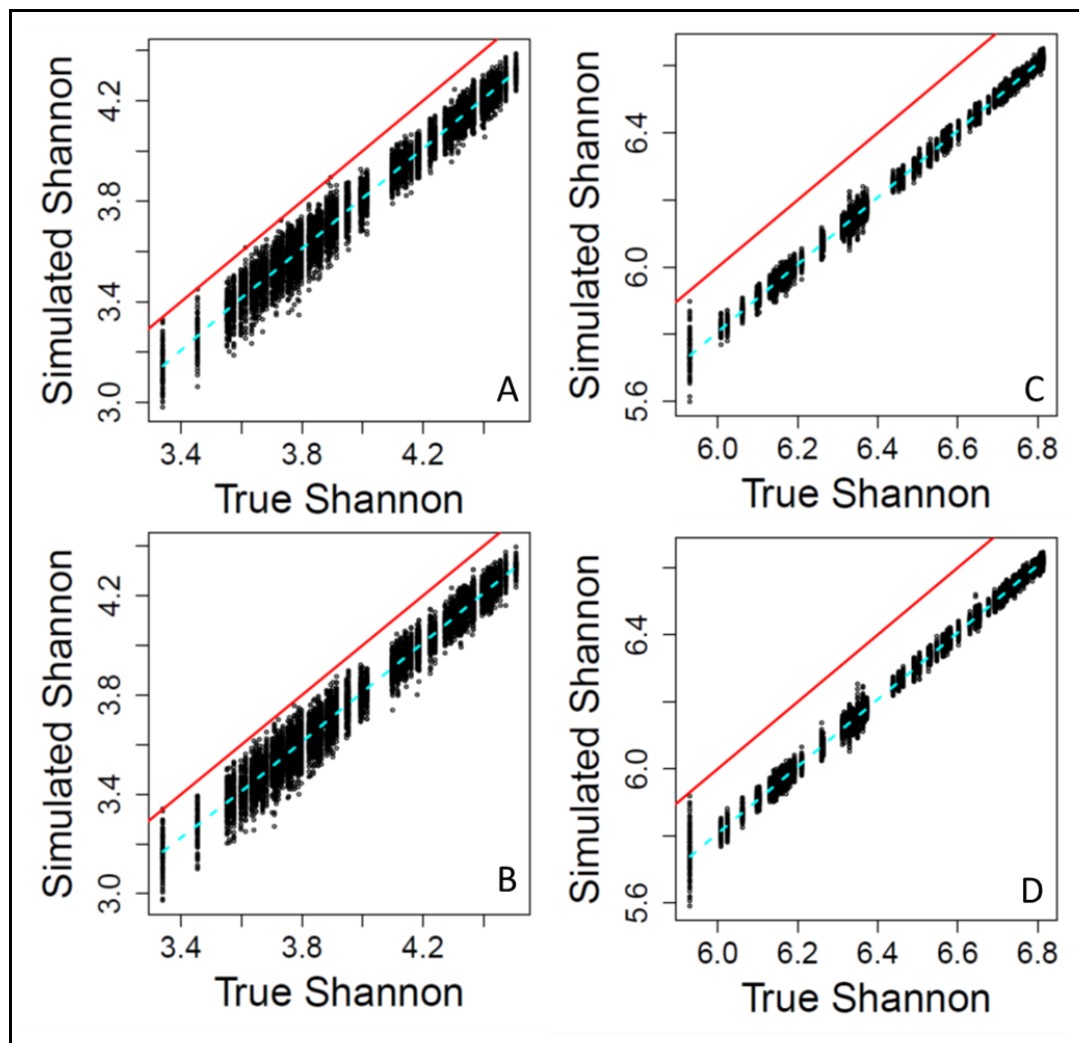

**Figure 9. Shannon α-diversity that includes simulated error regressed against true Shannon, across different scenarios.** (A) The same error applied to a given peak between samples, and 100 peaks per sample. (B) Different errors applied to a given peak between samples, and 100 peaks per sample. (C) The same error applied to a given peak between samples, and 1000 peaks per sample. (D) Different errors applied to a given peak between samples, and 1000 peaks per sample. On all panels the red line represents the one-to-one line and the dashed line is a spline fit to the data. All data are from the simulation model.




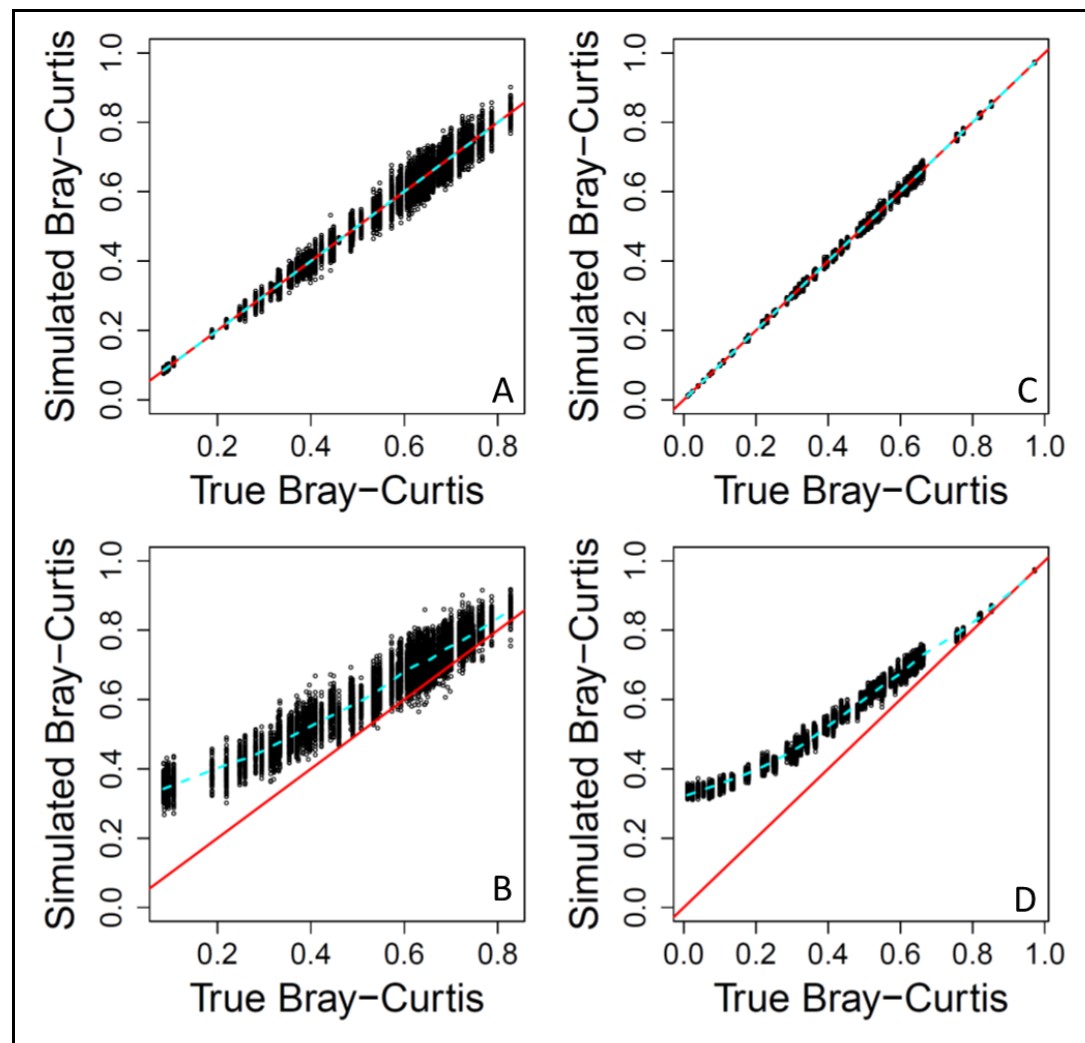

**Figure 10. Bray-Curtis dissimilarity as a measure of β-diversity that includes simulated error regressed against true Bray-Curtis, across different scenarios.** (A) The same error applied to a given peak between samples, and 100 peaks per sample. (B) Different errors applied to a given peak between samples, and 100 peaks per sample. (C) The same error applied to a given peak between samples, and 1000 peaks per sample. (D) Different errors applied to a given peak between samples, and 1000 peaks per sample. On all panels the red line represents the one-to-one line and the dashed line is a spline fit to the data. All data are from the simulation model.





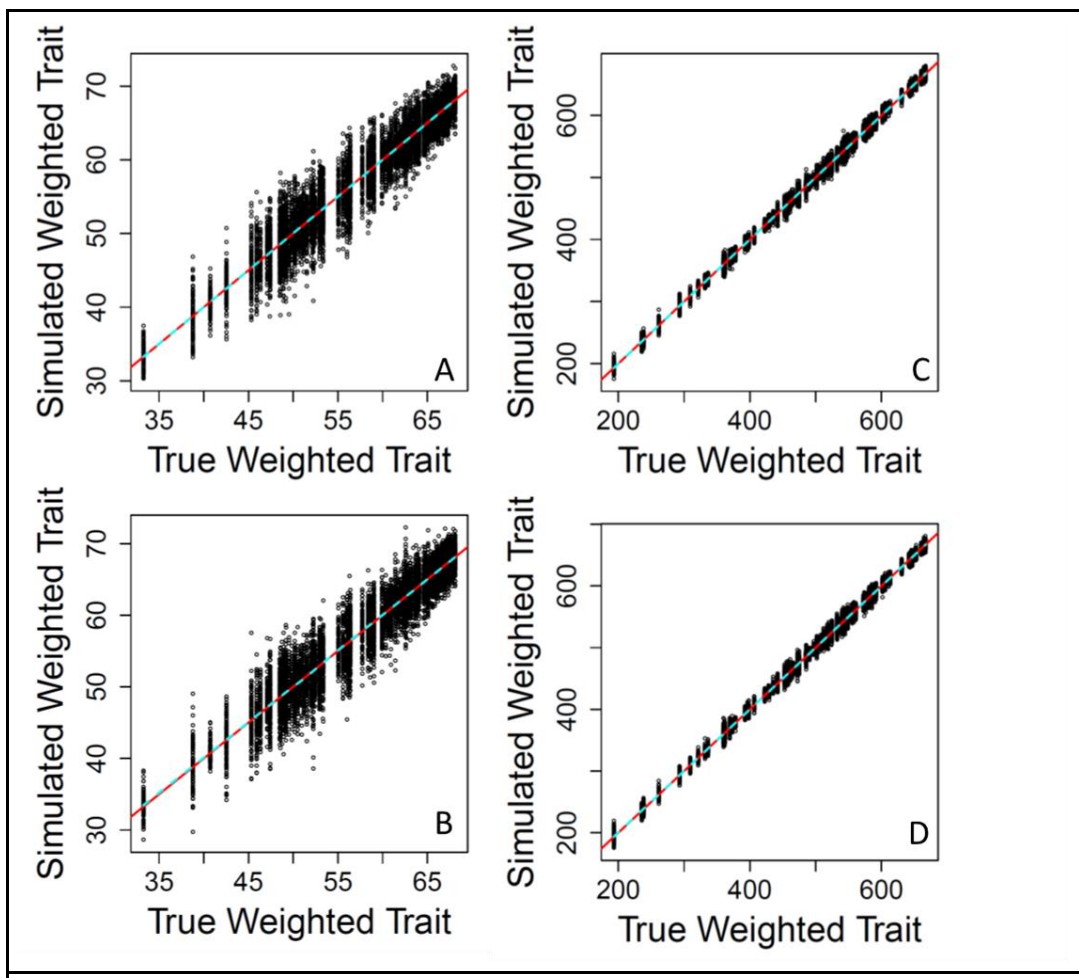

**Figure 11. Mean peak-intensity-weighted trait values that include simulated error regressed against true mean peak-intensity-weighted trait values, across different scenarios.** (A) The same error applied to a given peak between samples, and 100 peaks per sample. (B) Different errors applied to a given peak between samples, and 100 peaks per sample. (C) The same error applied to a given peak between samples, and 1000 peaks per sample. (D) Different errors applied to a given peak between samples, and 1000 peaks per sample. On all panels the red line represents the one-to-one line and the dashed line is a spline fit to the data. All data are from the simulation model.
