# Peer review of "Reviews and syntheses: Use and misuse of peak intensities from high resolution mass spectrometry in organic matter studies: opportunities for robust usage"

_EGUsphere, 2022_

## Author Response (AR1)

Dear Dr. Middelburg,

Below please find a point-by-point summary of how we edited the manuscript in response to each of the reviewers' comments. Our responses are in bolded blue text. We thank you and the reviewers for careful evaluation. Addressing the comments has, in our view, significantly improved the manuscript. We look forward to your further evaluation.

Sincerely,
James Stegen, on behalf of all co-authors

############

**Reviewer 1:**

General comments

The authors present a theoretical review of factors affecting the relationship between absolute quantity of a compound vs. analytical response as measured by high resolution mass spectrometry, as well as an experiment showing the differences in response factor for standard compounds in (negative ion) ESI-MS when measured in different matrices. They then discuss the ramifications of the fact that equal quantities of different compounds in a sample can produce different mass peak intensities or that the same quantity of compound may have a variable response in different samples depending on matrix, for the treatment of HRMS data from DOM analysis with statistical approaches designed for population ecology. As my expertise is mostly in the area of MS I will focus most of my comments on those sections.

The theoretical factors governing the response and identification of compounds in MS analysis, that the authors describe, are well known. Within the LC/MS community it is well known that one should not compare apples with pears, and that even comparing apples may be tricky as quantitation is difficult and often semi-quantitative. Not understanding the confounding factors can lead to over-interpretation of lc/ms data. Whether this message is something that needs to (still) be learned within the DOM community is unclear to me. I hope a reviewer from that community has more to say about that.

**Thank you for these comments. While we agree that quantitation has been discussed within the context of LC-MS, our survey of direct infusion DOM literature indicates a need to renew this discussion with respect to the analysis and interpretation of direct infusion FTICR-MS data. We plan to maintain the vision of the manuscript, as our literature review indicates there is a need for increased awareness of challenges and pitfalls across the DOM community. We observed that ~10 years ago, DOM manuscripts often included discussion points related to the issues of using peak intensities. More recent DOM manuscripts rarely mention or evaluate these issues, even if they are comparing apples to pears. We believe the current state of the field, in which methods using peak intensities have been used in previous papers and authors are picking up those methods**

**and applying them without necessarily understanding the pitfalls, necessitates the comprehensive treatment presented in our manuscript. In revision we did, however, significantly reduce the length of text across the sections discussing theory and empirical evaluation of peak intensities from mass spectrometry.**

The experiments with a set of standard compounds measured in different matrices is a nice illustration of the effect of ion suppression by matrix but is probably not necessary for the message of the manuscript as it is mostly a text book experiment with accordingly predictable outcome.

**We prefer to keep these analyses because while they may be textbook for mass spectrometry experts, we believe that many folks using high resolution mass spectrometry (HRMS) data would still benefit from these examples, particularly users of HRMS data whose formal training is in other disciplines such as ecology or biogeochemistry. In revision we did, however, significantly reduce the length of text across the sections discussing theory and empirical evaluation of peak intensities from mass spectrometry.**

The more novel part of this manuscript lies in the discussion on how these quantitative errors and uncertainties might influence the data treatment and outcomes. However, as I indicate in the comments below, I don't think that the in silico data set was generated with sound choices.

**Please see our comment below related to modifications to the *in silico* simulations.**

Below I list several specific comments and questions to be addressed. In general my recommendation is to shorten and simplify the descriptions of the factors governing quantitative response in (LC)MS, and focus the manuscript on the consequences for data treatment. A welcome additions would be a discussion on how to remedy the problems. As this probably entails more than major revisions, I recommend rejection at this time.

**While we agree with many of the suggestions from the reviewer, we believe that the recommended major structural revisions are related to the interpretation of LC-MS data rather than the focus of our manuscript, which is direct infusion (FTICR)-MS data. We believe that many of the sections regarding quantitative response which the reviewer recommends removing or shortening would directly benefit the FTICR-MS community. Thus, we prefer to maintain the overall manuscript structure. In revision we did, however, significantly reduce the length of text across the sections discussing theory and empirical evaluation of peak intensities from mass spectrometry.**

**Many HRMS data users have limited formal training in mass spectrometry. Our manuscript provides an overview of the issues to be aware of as well as an immediate solution in the form of a simulation model. We also provide ideas for additional solutions (e.g., hierarchical modeling).**

**We will also note that the other reviewer (Reviewer 2) does not necessarily agree with all our interpretations of the experimental data. This demonstrates a clear need for further discussion to reconcile how data are interpreted. In other words, what may seem "textbook" to some (e.g., Reviewer 1) is not so simple to others. The only way to advance the field is direct sharing, discussing, and improving research outcomes via primary literature.**

**We also emphasize that the manuscript is not focused on LC-MS data. We are focused on direct infusion (FTICR)-MS data. We edited the manuscript to make this point clearer, especially in the early parts of the paper.**

Specific comments

The second paragraph (lines 49 to 55) can be shortened and incorporated in the third paragraph, specifically in between line 67 and 68.

**We feel there are important details in text and prefer to retain this material. Other parts of the manuscript have been significantly shortened, however, such that the overall manuscript is far more concise.**

Line 75 to 76: peak intensity actually is proportional to differences in concentration for a certain compound, if the conditions are kept the same, otherwise no-one would be able to produce a standard curve. However the response factor (response per amount) may differ from compound to compound and for a compound depending on matrix and other factors. This should be rephrased.

**We revised this text to clarify that we are talking about the relationship between peak intensity and concentration in complex, natural organic matter samples.**

Line 96 to 99: move this section to the end of this section 2.3 as this is a concluding remark.

**This text was removed during the streamlining of the associated section.**

Line 101 to 112: Formation of (mixed) dimers and trimers should also be mentioned, as well as in-source fragmentation. There are several nice reviews on ion suppression , it would be good to reference a few here. The influence of the choice of ionization mode ( + vs - ionization; APCI vs ESI) should also be discussed. Also, a discussion on the use or (mis) use of internal standards would be a nice addition to this discussion

**We have shortened this section of text and removed some specific details on different ion type formations with ESI, and so discussion on formation of clusters is no longer within the scope of this text. We agree that different ionization modes will also have impacts, however our main focus in this text is a general discussion of the most common technique. We have added a sentence explicitly stating that experiment-specific**

**considerations (sample preparation, ionization mode, instrument specific parameters) are outside the scope of our focus. We refer the reader to a recent review discussion on that topic. (10.1021/acs.est.1c01135)**
**We also have added a reference to a detailed review in matrix effects in LCMS (10.1002/mas.20298) highlighting that these effects are only exacerbated in direct infusion workflows.**

Line 114 to 122: the issue described here is in fact not an ionization bias, but the inability to separate isobaric species. It simply describes the fact that when analyzing a complex (DOM) sample, any peak in any MS1 spectrum can in fact consist of the signal of multiple compounds with identical m/z (within the mass accuracy specifications) and is therefore a cumulative response of those compounds. This would still be the case even if the ionization of each compound was perfect at 100% efficiency.

**We changed the title of this subsection to 'Ionization Efficiency and Isomers" and streamlined the text.**

Section 2.3 The effect of dilution of any compound by abundant matrix should be discussed: in trapping instruments the fill rate of the trap (or ion target) is often a programmable parameter. If a compound, present at a given quantity is the most abundant compound there, than the trap is mostly filled with that compound. If the compound is present in the same quantity but with together with a lot of matrix ions, the matrix ions will take up part or most of the available space in the trap, effectively 'diluting" the compound and thereby to underestimation of its actual quantity. Maybe a better title for this section would be "ion transmission and collection"

**We changed the subsection title to that suggested by the reviewer and added the following sentence: "Increases in the true abundance of other ions can decrease the measured peak intensity of a given ion due to a dilution effect resulting from a finite number of ions that can fit within the ion trap."**

Lines 172 to 251: Section 3, in my opinion, can be removed in its entirety. These are predictable text book experiments and whatever extra information is discussed can be incorporated in section 2.

**As discussed in response to earlier reviewer comments, we strongly prefer to retain this section. While mass spectrometrists may not be surprised by results shown in this section, the intended audience for this paper is much broader. We posit that the average ecology- or biogeochemistry-minded data user will not necessarily be aware of the issues or patterns addressed in this section. A key goal of our paper is to speak to researchers across the continuum, from ecologists to mass spectrometrists. At both ends of that continuum, researchers will find some of our examples very simple and 'textbook' in direct relation to their domain of training. However, few researchers are trained in all the concepts covered in our paper, and thus the paper provides common**

**ground for anyone working with direct infusion mass spectrometry, regardless of formal training domains.**

Line 331 to 354: generation of in silico data set. I find the use of errors well above 1 (like 1.5) debatable. Although ion enhancement does happen it is very rare and seldom that pronounced. Even in fig 4, the increase shown in panel C from 0 to 2 ppm matrix added, is attributed to the addition of endogenous compounds from the matrix. So is this realistic? What would happen if you make the error non-gaussian?

**We understand the questions raised by the reviewer and have made one targeted change to the simulation model; we modified the error range to more closely match the empirical data. This had no influence on the simulation outcomes; we have included a set of supplemental figures to show this and the code and figures are all in the publicly available GitHub repository.**

**More generally, we acknowledge that 1) there are multiple ways to set up the simulations, 2) we expect future developments in this vein, and 3) we are quite sure that different researchers will suggest and prefer different implementations of multiple aspects of the model. Part of the reason that many valid setups exist is that we are simulating phenomena that are not deeply characterized and may never be truly known.  However, our simulation model's primary value lies not in the particular conditions simulated in our paper, but rather in its ability to be adjusted to more closely match a dataset of interest, thus providing a flexible framework by which mass spectrometry users can evaluate their own data. To this end, we provided the code used to produce the simulation model in a publicly available format.**

**We also edited the text to more clearly highlight the general utility of the model in the revised manuscript. A great follow-up manuscript would be a deep exploration of how the ecological metrics respond to different assumptions in the model, as well as extending to additional ecological metrics as pointed to in the manuscript.**

Line 375: Again a comment about the choices underlying the in silico data set: Ionization efficiencies do not randomly vary for a compound across samples. In general, more complex samples with more matrix will have less ionization efficiency for all compounds in that sample. The deviations are not truly random.

**In summary, we edited the manuscript to include additional ideas for simulation model development, such as the idea suggested here. Below, we provide more rationale for this plan, as opposed to myriad sensitivity analyses.**

**As noted in the manuscript, the presented simulation model is intended as a first look at what biases and/or issues may arise when peak intensities are used to calculate common ecological metrics. We feel it is important to keep the scope of the modeling as constrained as possible while still providing informative outcomes. While the point made**

**here by the reviewer is interesting, the practical question is whether we should expand the scope of the simulation model to account for it, or if we should provide this in the manuscript text as an example of future development needs.**

**More specifically, we ran the model with either ~100 or ~1000 peaks within each sample. The more peaks in a sample is, we believe, what the reviewer refers to as 'more matrix.' In this case, we could make ionization efficiency of all peaks inversely proportional to the number of peaks in a sample. We believe that this change will have a limited impact on the ecological metrics because all samples in a given simulation run have about the same number of peaks. Where this change could have more influence is in comparisons across samples that vary widely in the number of peaks (e.g., comparing one sample with 100 peaks vs. one sample with 10000 peaks). In theory, we could pursue this direction and implement a function that decreases ionization efficiency for all peaks in a sample based on how many peaks are in that sample. However, the extent to which this function accurately models physical phenomena is unclear, and this change will also add a significant amount of scope to the simulation component of the manuscript. For instance, we would need to add another large set of figures across both of our current error scenarios and include sensitivity analyses for how the effect of this mechanism changes depending on details of the function.**

**We are reluctant to expand the simulation scope in this way, which will add numerous figures and substantial length to the (already long) manuscript. In addition, we are not fully convinced that more peaks in a sample will, by itself, decrease ionization efficiency. Peak intensities may decrease, but that is because the concentration of each peak/molecule must be lower if there are more kinds of peaks/molecules and total dissolved organic carbon concentration is kept constant. Decreasing peak intensity due to lower concentration is not the same as decreasing ionization efficiency.**

**Given the above considerations, we added some new text in the manuscript laying out additional ideas for simulation model development, such as that suggested here. The added text is in the Conclusion section and reads: "The model should be expanded by including additional ecological metrics/analyses, more than two-sample situations, sample-to-sample variation in peak richness, links between peak richness and peak intensity, other ways of modeling error, and measured levels of error between concentrations and peak intensities."**

Lines 385-408: as wordy as some of the other section are, as quickly the authors go through the consequences for the application of the statistical models. Some parts were truly unreadable to me as a person not involved in statistics and models. The concept of mean trait values and figure 11 are hardly introduced and poorly explained.

**Our inference is that the primary challenge here is with the mean trait values. In turn, we add additional explanation that reads: "We also assigned an arbitrary trait value to each peak and calculated true and observed sample-level mean trait values; the mean values**

**for each sample were weighted by true abundance (true mean) or observed peak intensity (observed mean). This is analogous to the commonly used approach in ecological studies of computing community-level abundance-weighted trait values, such as plant leaf area index or animal body size (Muscarella and Uriarte, 2016). This approach is also common with HRMS data, such as sample-level peak-intensity-weighted values of hydrogen-to-carbon ratios and molecular weight (Roth et al., 2019; Wen et al., 2021)."**

Line 483 – 489; After reading the entire manuscript the conclusion that the ecological metrics actually perform quite well, came as a bit of surprise to me. The tone of the manuscript as a whole is quite negative towards these concepts.

**Based on this suggestion, as well as similar comments from the other reviewer, we edited the text throughout the manuscript to have a more positive and future-looking tone.**

Fig 4: What does relative intensity mean here? Relative to what? As no bar is ever reaching 100%, is the response of the compound with no matrix or SRFA added 100%? Please clarify. Which of the differences between bars is statistically significant? However, I recommend to remove this figure along with section 3.

**As discussed above, we greatly prefer to retain this section and this figure. By 'relative intensity,' we mean that the data were scaled to the largest signal in any replicate from that series of spectra. We are combining replicates to show the mean values with a 95% confidence interval, which is why the bars do not always reach 100%. These details have been added to the manuscript by editing the Figure 4 caption, the front end of which reads: "A) Barplot visualization of the relationship between signal intensity (relative intensity) and concentration of analyte for three chemically distinct molecules analyzed contemporaneously but independently in pure methanol solvent. Relative intensity indicates data were scaled to the largest signal in any replicate from the associated series of spectra. Replicates are combined to show their mean and 95% confidence interval."**

**The primary value of the figure is to show how observed peak intensities deviate from expectations and assumptions made by ecological metrics commonly applied to direct infusion mass spectrometry data. As such, we prefer to not add details of statistical significance, beyond providing the 95% confidence intervals, as adding this information would introduce significant complexity to the existing figure, likely contributing to more confusion than clarity.**

Fig. 8. Panel A, B and D are data clouds, which seems a logical outcome of the way errors are assigned in the in silico data set. But why the convergence to 0 for the true to observed difference in panel C? I do not see the strong resemblance between panel A and C described in line 383 of the text.

**We added some explanation for why the data converged to zero in the middle of the data cloud. The following text was added to section 5: "In Figure 8B the differences collapse when near zero because when two samples have essentially the same peak intensity for a given peak, introducing the same error to that peak in both samples has little influence on the between-sample difference in peak intensity."**

**The point about lack of resemblance is due to an error whereby figure labels were misplaced (i.e., panel B should be C and vice versa); we corrected this error and thank the reviewer for catching it.**

Citation: https://doi.org/10.5194/egusphere-2022-1105-RC1

**Reviewer 2:**

Given the growing literature on the application of ecological analyses to high-resolution mass spectrometry, a careful assessment of the bias, flaws, and assumptions is timely and a welcome contribution to the expanding subdiscipline. In particular, the suggestion to expand modeling approaches, including machine learning or hierarchical modeling, to quantify the magnitude of errors is an important one for future research.

**Thank you for the encouraging remarks.**

Meanwhile, the empirical results in this study show that many ecological metrics derived from the peak intensities provide valid patterns. This is an important result and could be highlighted alongside papers demonstrating the acquisition of quantitative data from HMRS (e.g., Kruve et al., 2020; Groff et al., 2022).

**We added a reference to Kruve 2020 which highlights strategies for quantitative analysis of non-targeted LC-MS workflows. This was added in section 2.1, and the revised text reads: "Ionization suppression can be mitigated by online separation whereby non-targeted LC-MS approaches may yield more quantitative data (Kruve, 2020), but matrix effects remain a significant issue even for LC-MS (Trufelli et al., 2011)." Kruve 2020 and Groff et al. 2022 are both focused on LC-MS workflows, however. Our manuscript is contextualized around direct infusion analysis which lacks online separation. As such, we also edited the manuscript throughout to clarify that the focus of our manuscript is on direct infusion analysis.**

On line 483, it is stated that HRMS has many weaknesses, just like any analytical platform. Most practitioners of HRMS would agree that the biases presented in this paper are present (as reviewed in Urban et al., 2016; Kujawinski et al., 2010). Many of these biases (Viera-Silva et al., 2019) exist with other compositional data as well (unlike the statement on lines 73-77), such as microbiome data and have produced solutions such as those reviewed in Gloor et al 2017, such as the creation of internal standards (Hardwick et al., 2018).

**We added references to the suggested papers and pointed out there are methods developed in other fields that could be helpful for FTMS data. That text is in the final paragraph and reads: "In summary, FTMS has many strengths and weaknesses just like any analytical platform. Other types of compositional data also contain biases and uncertainties, such as the lack of true quantitation in sequence-based microbiome data (Gloor et al., 2017). Careful use of FTMS peak intensity data informed by objective, model-based guidance can overcome some of its weaknesses. We encourage further development of the model presented here and inclusion of additional methods developed to address issues that arise in similar data types (e.g., Gloor et al., 2017; Hardwick et al., 2018; Vieira-Silva et al., 2019)."**

**We also edited the text around lines 73-77 to clarify that we are referring specifically to FTMS data. That text now reads: "These studies may be discarding useful information, though it is unclear what biases and uncertainties are introduced into ecological metrics when using FTMS peak intensities."**

Together, these two foci highlight the most problematic aspect of this paper: The tone is much too negative to accurately reflect the reality of the (very common) use of peak intensities. A more balanced and contrasted view should be taken so as not to alienate specialist readers or mislead those less well-versed. The tone leaves the feeling that the bulk of the literature in the past decade is highly flawed and not to be trusted. This is not impossible, but if this is what the authors are trying to convey, the analysis must be made much more robust. I suggest the authors change the tone to ensure that the key messages (the utility of ecological metrics and some of their drawbacks) are most effectively conveyed.

**The other reviewer had a similar comment. We did a thorough edit of the whole manuscript to present a more positive and forward looking tone.**

I am also concerned that some of the assumptions of the empirical work have significant technical flaws. This may be improved by providing greater transparency for the selection choices.

**Please see comments below regarding our choices of compounds to empirically study.**

1) The errors of their simulation model.  How can a random selection between 0 and 100 for the simulated errors be justified (lines 352 and 369)? Why is 0 included in the random selection? The decision for this range should be motivated by actual evidence, such as from the experiments measuring variation in peak intensities of analytes of known concentration. Examining Fig, 4b. it looks like a better error selection would be between 1-8. Without further justification, the results of the in silica simulation model appear quite arbitrary.

**The reason for including zero is to allow molecules to drop below the limit of detection of the instrument, such as in the cases of very poor ionization given either their inherent chemistry and/or interactions with other molecules in the sample/matrix. To address the**

**question about sensitivity of the simulation outcomes to the range error range, we used the empirical data (as suggested) and re-ran the simulations using an error range from 0 to 8. This had no influence on the results. The outcomes are in the GitHub repository and are included as supplemental figures.**

**The revised text is within section 5 and reads: "The inclusion of 0 indicates situations in which a given peak (i.e., ion) does not ionize well enough to be observed. The results should not be sensitive to the selected range, but as a sensitivity analysis we also used a distribution of errors ranging from 0 to 8. This narrower range is suggested by our empirical data (Fig. 4B), but simulation results were not affected (Supplementary Figs. S3-S8)."**

2) Peak intensities are normally distributed in HRMS data (e.g. He et al., 2020). The way the authors generate random intensities does not reproduce the normal distribution in peak intensities.

**Typical FTMS spectra of NOM, especially highly processed standards like SRFA, show an envelope of peaks across the m/z domain which looks similar to a normal distribution. However, peak *intensities* are not normally distributed. That is, if you take all the peak intensities and sort them by intensity - rather than by m/z - the distribution has an apex at low intensities, and is closer to a poisson distribution. Further, NOM samples are typically more heterogeneous than SRFA due to a higher mix of 'fresh' organic matter components. Given this heterogeneity, we believe our approach to random sampling is appropriate. We have also made recommendations in the conclusions section that the modeling approach can be expanded for more experiment-specific analysis.**

**We also edited the text to highlight the possibility of modifying the simulation model to reflect different distributions of peak intensities. The text in section 6 now reads: "It should be possible to include the number of samples, the number of peaks in each sample, the peak intensity distributions, number of replicates, and the specific ecological analyses that will be applied."**

3) The number of peaks.  The simulation models use either 100 or 1000 peaks.  These are not environmentally relevant.  Most environmental studies have several thousand peaks, where the authors nicely and unequivocally show in Fig. S2 that there is absolutely no bias in the calculation of ecological metrics, that is, the observed vs true $R^2$ values approximate 1.  For this reason, any simulations with small numbers of peaks are misleading and not relevant to most studies.

**With respect to the 100 and 1000 peak simulations, we note that real datasets vary tremendously in the number of peaks being used for analyses. There are many reasons for large variation in the number of peaks actually used in the calculation of ecological metrics. For instance, some researchers may only examine peaks with assigned formulas and/or peaks observed consistently across technical replicates.  It is important for**

**researchers to be aware that biases and uncertainty will increase as the number of peaks used decreases. Thus we feel that the continued inclusion of the analyses, figures, and discussion informed by simulations with 100 or 1000 peaks is warranted.**

**We edited text at the end of section 5 to read: "The variation in observed values explained by true values (via a linear model) increases rapidly with the number of peaks, and sharply asymptotes beyond ~500-1000 peaks per sample (Fig. S2). Sample-to-sample changes in the value of ecological metrics can, therefore, be interpreted with increasing confidence as the number of peaks increases. Qualitative gradients are, therefore, more robust with more peaks. The absolute magnitude of some ecological metrics, however, are shifted away from their true magnitude even when there are large numbers of peaks (e.g., Fig. 10D). Quantitative comparisons from one dataset to another may, therefore, require further simulation-based evaluation. We also caution that the number of peaks needed to reach the asymptote, thereby minimizing error, is likely dataset dependent and 500-1000 peaks should not be taken as a general rule for real-world datasets. We encourage researchers to complete such simulations using the numbers of peaks present across their real-world datasets to better understand their ability to make statistical and conceptual inferences."**

4) Why these specific standards? What are their features beyond just an absence in natural DOM.  I think there needs to be a description of what makes the molecular structures, ionization properties, etc… of these analytes appropriate spike-ins?

**We added a sentence in section 3.1 to explain the rationale behind these standards. The sentence reads: "We selected chemical standards which are natural products with molecular formula and chemistries typical of compounds commonly observed in organic matter, and were amenable to negative mode ESI analysis."**

5) No evidence is presented why these higher analyte concentrations (>200 ppb) or with relative intensities of individual peaks >1% will ever be realistic.  I similarly don't understand why the summed relative intensities exceed 100% in Fig 4a in the absence of SRFA.

**We agree that the higher concentrations we used may be greater than typical concentrations for individual endogenous molecules in NOM, but again note that in a typical NOM mass spectrum, any given peak is the sum of many different isomeric compounds. By using individual compounds, we have to increase their concentration to compensate for this reduced complexity. We have text to this effect in section 3.1, which reads: "These standards were analyzed at higher concentrations than typically observed for NOM because they were single compounds rather than formula-summed features (with multiple isomers) within a NOM spectrum; higher concentrations were required to compensate for lower isomeric diversity."**

**We clarified the relative intensity scale in the figure caption. Specifically, we explain that 'relative intensity' means 'scaled to the tallest feature in the spectrum'. _i.e.,_ the base peak**

**is at 100%, and other peaks are relative to that. This is why the sum of relative intensities exceeds 100%. Further, the panels C,D,E are all on the same scale as each other - which highlights the signal suppression from using a BondElut matrix rather than pure methanol alone.**

There are also several strong statements that I do not believe are sufficiently supported by the scientific evidence presented in the paper. These are on line 245 "Strategies to use calibration curves will fail" and line 324 "The previous sections show that between-peak changes in peak intensity do not accurately reflect between-peak changes in abundance".

**As part of our thorough revision to the tone of the paper, we revised these statements to be more positive and forward looking. With respect to line 245 in the original manuscript, we edited the end of section 3.2 to read: "Combining the empirical results from this subsection and the previous subsection with instrument theory discussed above suggests significant uncertainty in relationships between true concentrations and peak intensities from direct infusion FTICR-MS. Calibration curves can be used in the simplest of situations, but may be challenging when there are structural isomers and sample-to-sample variation in matrix composition. Modeling of constrained systems may, however, allow for data-driven and mechanistic data normalization strategies for enhanced use of peak intensity data."**

**With respect to line 324 in the original manuscript we edited the start of section 5 to read: "The previous sections highlight challenges in connecting between-peak changes in peak intensity to between-peak changes in abundance (Fig. 4)."**

**We also note that Reviewer 1 appears to agree with our interpretations and feels that the data shown are too basic to warrant publication. They suggest removing an entire section of the manuscript because it is 'textbook.' Reviewer 2 interprets the data differently, however, which indicates there is variation in how researchers view and use peak intensity data from direct infusion FTMS. This emphasizes the importance of retaining all sections of the manuscript and more generally to continue publishing the type of empirical studies we provide to further the literature-based discussion.**

Fig. 4 seems to show that the relative intensity scales with concentration. The authors can predict this by a nonlinear model or GAM for each analyte, or with a single model where the slope varies with the m/z (for example). Without having attempted such an analysis it is difficult to understand how this statement is supported.

**Figure 4 does show that for an individual, known compound, matrix-matched, dilution ladder, concentration is proportional to signal intensity. This is indeed how quantitative mass spectrometry approaches work (i.e., through calibration curves). However, those calibration curves are molecule-specific and dependent on matrix effects. This is evidenced by the changed behaviors in Fig. 4D and 4E, relative to 4C. Thus, it is not possible to generalize such calibration curves when considering NOM samples**

**containing thousands of different unknown molecules with unknown isomeric complexity. Fortunately, our simulations show that it may not be necessary to model or otherwise explicitly derive quantitative relationships between intensity and concentration because the ecological metrics perform well without that information.**

Figure 4 also nicely shows that you can reach quantitative assumptions between-peaks. As shown in Figure 4a, at low concentrations of the three different molecules (<100 ppb), the signal intensities seem statistically indistinguishable. A similar result is seen, especially in the MeOH Matrix, at higher concentrations of SRFA that effectively dilute the analytes to representative concentrations. These results suggest that the analytes are performing quantitatively.

**While this is an interesting observation by the reviewer, we do not feel it is generally applicable in complex mixture data. Our data show that there is inconsistency in how well differences in peak intensity map to differences in concentration. It may be that there is a clearer link at low concentration, but in natural samples that vary widely in complexity, freshness, etc., the true concentrations of each peak are unknown. Put another way, one could not tell if a low-signal ion is due to a high-concentration analyte with poor ionization efficiency, or due to a low concentration analyte with high ionization efficiency, nor if a high-signal ion consisted of a single high concentration analyte or many low-concentration isomeric analytes. Thus, we have no way of knowing which peaks have intensities that carry valid information and those that do not. As such, we cannot generally recommend using between-peak differences in intensity to infer between-peak shifts in concentration.**

**We added text to section 3.1 to help address this point, which reads: "We note that absolute differences in signal intensity may be smaller between molecules at lower concentrations, but this does not necessarily mean that low intensity signals consistently indicate low concentrations and this does not aid in quantitatively interpreting higher intensity signals."**

**More generally, we again highlight the contrast between the two reviewers on these topics. Reviewer 1 feels our empirical data show outcomes that are 'textbook' and thus are not needed because it is well-known that peak intensities cannot be used quantitatively. Reviewer 2 feels our data support the quantitative use of peak intensities. Clearly, there are diverging opinions across the research community, highlighting the importance of this section of the paper. We need to come to a resolution on these topics as a research community and that can only happen by presenting, discussing, and improving evidence.**

Additional comments:

Line 60: The authors should cite the reference of the first use of 21T FT-ICR-MS (Smith et al., 2018).

**The citation has been added.**

L195-205 – This point is already made on L122

**The text has been streamlined to minimize length and duplication.**

Figure 4- relative intensity is not explained. Relative to what?

**Text has been edited to clarify this (see comments above).**

---

## Referee Report (RR1)

General comments:

This study mainly showed that the peak intensity of FTICRMS is not a proxy for abundance and concentration. However, this is somehow common sense and should be already known to most researchers. The experiments performed by the authors nicely proved that peak intensity should be used with caution, especially for quantification. Again it is already well known. The author claimed that there are plenty of misuse studies of peak intensity, if this is true, I would suggest the author organize a table of literature to illustrate how serious this issue is. The simulation model is great but I highly doubt its applicability considering so many speculation factors were introduced, including random error, etc. Besides, this model was to evaluate the matrix effect on peak intensity, and seems that it can only be used as an educational tool for people to understand the bias of FTRCRMS in the quantitative study. How people can use it for their own samples? For the model part, I suggest the author simplify the sentences as it is hard for me to understand them and, if necessary, list the mathematical equations used for calculation. Add text to clearly and concisely describe what this simulation model is, how it can be applied to environmental samples, and how people can make use of it. This is very important because the model is the only highlight I can see from the manuscript even though it is very hard for me to understand all. I suggest a major revision for this manuscript and the author should pay attention to the concise of writing since the authors aim at people who are new in this field. For specific comments see below please:

The introduction part mainly described the diversity studies (yes or no question), then all of a sudden in the last paragraph, raised the concern of "quantification" (how much question), What is your point here? Could the diversity part be removed and more focused on how people used the peak intensity data incorrectly to support the concern of the authors?

The study lacks a "Method" part, including information on compounds, standards, instruments, etc. Please provide.

Sections 3.1 and 3.2 should be improved, use a scatter plot for Fig.4 instead so that you can simplify the text and make it easy for readers.

It seems that the homogeneity of the sample sets is not taken into consideration. Real-world within peak comparison might be more complicated if the studied set contains very heterogeneous samples.

L75: There was only one paper cited, but later sentence uses "these studies", add more papers please.

L113: "Using consistent sample concentrations" is impossible

L145-L146: Add text to describe what samples, what compounds, and what chemical standards.

L268-L269, list the publications that misuse peak intensities

L369: add literatures

L275-L276, why do you choose 100 and 1000 peaks? Where are these peaks from? From what kind of samples?

Figures: Be consistent, either use figure and Fig, do not mix the usage

Fig.4 Please change it to a scatter plot and give a correlation value (R2), this can help people understand the relationship between intensity and concentration. In fact, in lines 162-163, the author also suggested a calibration curve, why not do it for this study?  What is relative intensity?

Fig. 8: How do you get the observed difference? How is the true difference calculated? The R2 in Figure 8C is questionable: although the value appears to be high, it's obvious that the independent variables are heteroscedastic.

---

## Author Response (AR2)

Dear Dr. Middelburg,

Thank you for securing two helpful reviews of our work. Below we summarize the revisions we made in response to the reviewer suggestions. Reviewer suggestions are in normal text and our responses are in bold text to easily differentiate. We feel the reviews were helpful in improving the manuscript. We look forward to your further evaluation.

Thank you,
James Stegen on behalf of all co-authors

######

Reviewer 1:

In this manuscript Kew et al. discuss the use of intensity values for natural organic matter ions measured by ultrahigh resolution mass spectrometry. The authors review basic concepts, provide a summary of limitations, pitfalls, and then describe how the use/misuse of intensities could impact the employment of FTMS to obtain ecological metrics. The paper is concluded with a wonderful summary and recommendations to the community. The paper is not of a traditional format and is a hybrid between a review article, a critique, has some experimental data, as well as some computational modeling. It also has a lot of important citations provided to us, which is also very valuable. As a heavy FTMS user I enjoyed reading through and will be certainly coming back to in when I need to be reminded of how intensities can change if I were to change X, Y, or Z in my experiment.

**Thank you for the encouraging remarks, we're glad to hear you see value in this work, especially as a heavy user of FTMS.**

I really like figures 1,2, and 3. I agree with the authors, as they say in the response to comments doc, that most environmental users of FTMS do not have much formal education in mass spectrometry and they end up learning MS and other analytical methods "on the go". So while some of this theoretical background and experiments may be redundant with other papers/textbooks, often such resources are abundant with complex diagrams, lots of calculus, etc. - texts that are not super digestible by the broader biogeoscientist. This study is a perfect resource for our community and especially the uprising young scientists in non-chemistry departments (e.g., geology). The authors also nicely introduce the ecological metrics to non-ecologist readers (as a classically trained chemist I have no formal education in ecology, so it was nice to read a broader introduction of alpha/beta diversity indices and their uses).

**Thank you for the further encouraging remarks, we're glad you find value in the introductory elements.**

Given that this is a second round of review, this is already an excellent paper. I have one major comment and a few minor ones that can be easily remedied with a minor revision and overall aim to make the paper more receptable by the broader biogeoscientific community.

My biggest critique is that there is too much emphasis on the ecological indices. The PNNL group is the main group using these metrics with very few other groups using them once in a while - if we look at the global biogeosciences community, most people do not use mass spec data to calculate these diversity indices and do not use the data in an ecological framework/manner. People commonly plot van Krevelens, DBE vs C, calculate %CHO, %CHON, %CHOS, etc. types of formulas (%lignin, %lipid, etc.), other averages (average H/C, O/C, NOSC, AImod, etc.). Statistics like HCA, PCoA, PCA, NMDS are also very common, and so are now Spearman correlations with some other metrics (PARAFAC Fmax, CDOM metrics, etc.). I am not saying this is the right thing to do, I do think people should be doing more advanced things with MS data (ecological frameworking, neural networks, etc.), but this is where we are at and have to consider the state of the community right now. As this paper is targeting the broader community, it should contain as much relevant information as possible.

**This comment helped us realize we needed to be more direct in the text that we include sample-level intensity-weighted trait/property/characteristic as one of our 'ecological metrics.' This is directly related to commonly used intensity-weighted averages (e.g., of H/C, etc.) that the reviewer points to above. The revised manuscript is more direct about including that as one of our three studied ecological metrics. In addition, we study Bray-Curtis dissimilarity, which is commonly used as input to NMDS and discuss the utility of Spearman rank-based correlation analyses. The revised manuscript is more direct about the relevance of the associated simulation-based analyses and outcomes to real-world FTMS NOM studies.**

I like the ecological framing, but I do not think it would be so useful to the broader biogeoscientific community, because most of us do not use this type of ecological framing for our experiments at present and probably in the foreseeable future. For this reason, I think sections 4 and 5 are too long and should be combined into one section. The ecological section should come after a main section where the more common uses are discussed. All of the things I describe above involve the use of intensity values and the authors should discuss the relevant issues.

**As noted in our response above, the ecological framing goes beyond the diversity metrics and is quite relevant to a broad range of common analyses. We elaborate further on this in our next response, below. The reviewer also suggests shortening sections 4 and 5. We significantly shortened section 5, which describes the simulation model and associated results. We moved more than half the text to the supplementary materials and also moved two multi-panel figures to the supplementary materials. The text remaining in section 5 is written to be more accessible to a broader readership, and we leave the technical details to the supplementary material. We greatly prefer to keep section 4 as is because it provides a conceptual summary of how to best think about FTMS data when**

**doing NOM studies through space and/or time. It is written to be accessible to a broad audience and is an important part of the storyline; we ask for support to retain it.**

Some examples I can easily come up with: Should we use intensity-weighed H/C, O/C, etc. metrics (sum of rel.intensity*metric), or just regularly calculated averages? Would it be better to report %CHO, CHON, etc. as % intensity (intensity of CHO formulas/all formulas) or % number of formulas (number of CHO formulas/number of total formulas)? When people look at a three-dimensional van Krevelen plot, and see that the most intense peaks are in the lipids region (for example), how confidently can we say "the sample is rich in lipids/most molecules are lipid-like". When people Spearman-correlate FTMS formula intensities with external metrics (e.g., proteinaceous component from PARAFAC), how much can we trust that the signifficantly correlating formulas correspond to proteins? Regarding statistics, should we combine FTMS results (%CHO, %CHON, etc., which are semi-quantitative) with truly quantitative data (DOC, SUVA, etc.) - everyone does it, even though all variables have to be of equal quantitativeness. Or should we not use FTMS intensities at all in such stats and just resort to presence-absense (see first paragraph of section 2.6.5 here: 10.1016/j.gca.2010.03.035). I can go on and on with coming up with similar questions, and I am sure that the authors can too - I don't necessaryly ask you to answer these exact questions in the revised version, but giving you ideas for talking points. I think this manuscript is the perfect place for addressing such questions and having this kind of discussion - this would be much more beneficial to the community.

**These are all great questions and our work addresses many of them. We included more discussion on these examples as part of our recommendations, within section 6 (the last part of the paper). We kept this after the simulation model because the simulation model results are key to generating our recommendations. As noted above, we cut the simulation model section (section 5) by more than half to help with readability and broad accessibility.**

After such section of describing the use of intensities in "common FTMS data workup approaches" you can follow up with the "using/misuing FTMS intensities in an ecological framerworks" section. I will also say that sections 4 and 5 are collectively too long, not so straightforward to read by someone who does not use ecological indicies on a regular basis (in my opinion, most of FTMS users), and so I recomment shorteing, not going that much into the weeds, and slightly streamlining with the thought of making that more friendly to the broader public.

**See above; in short, we took the advice of shortening section 5 significantly.**

A minor comment was that I was confused by the "within-peak" and "between-peak" terminologies. Regarding within-peak, I first thought of doing comparisons of the isomers that are under one m/z value. The between-peak sounds ackward and I though we would be comparing the noise level between two peaks.... I recommend changing these terms to something clearer. Suggestions: "same-peak comparisons" for within-peak and "peak-to-peak comparisons"/"different-peaks comparisons" for between-peak.

**The language used here is difficult to navigate and we feel that no choice is clearly better than all others. For example, if we use 'different-peaks' instead of 'between-peaks' we'll end up with some sentences reading '...different-peak differences...', which feels potentially confusing. We're also careful to define our use of within-peak and between-peak at the start of Section 2:**

**"Here, we define 'within-peak' as comparing peak intensities of the same feature (i.e., m/z or molecular formula) across different sample spectra and 'between-peak' as comparing peak intensities across different features."**

**To help with any confusion regarding isomers, we added the following sentence to the same paragraph: "Both within-peak and between-peak comparisons are fundamentally based on the m/z observed within a mass spectrum and neither address comparisons across isomers."**

**Figure 2 also provides a visual definition of within-peak and between-peak. Given all these considerations, we propose retaining these terms.**

This manuscript desperately needs to include some text (maybe one paragraph?) about normalization. Usually we take the raw intensity values of assigned peaks and divide the by the sum of all intensity values of assigned peaks, but is this the best way? Should we consider the sum of ALL spectral peaks (including isotopologues, blank peaks, etc.) or just normalize to the sum of intensities of the assigned formulas? Some ideas for talking points. I am familiar with this wonderful paper that provides some guidance: 10.1002/rcm.9068

**We appreciate this suggestion and in response we added a short paragraph within a new sub-section (3.3). We believe that post-hoc normalization strategies (such as detailed in the referenced paper) are helpful for some applications, but cannot mitigate the underlying physical processes that cause peak intensities to be weakly related to true abundance. We feel that deeper discussion on normalization is beyond scope of this manuscript primarily because normalization isn't a clear solution to the challenges we raise. Regardless, we agree it's helpful to call this out directly in the manuscript, as done in the new text (provided immediately below).**

**The following text was added:**
*"3.3 Data Normalization Strategies*
*In the previous section, we use the peak intensities for each analyte without any normalization, only scaling to the base peak or between spectra to make comparison easier. However, more sophisticated or comprehensive normalization strategies may be useful when trying to make quantitative inferences of the data. Considerations may include whether to use the total intensity within a spectrum (including noise, isotopologues, and unannotated features), or to use just the peak intensity apportioned to annotated features. Additionally, non-linear or more sophisticated functions may have benefits. Such post-hoc statistical approaches have utility for some applications but do not resolve the fundamental, underlying physical origins of the weak connection between peak intensities and true concentrations. We refer readers to the work of*

*Thompson et al. (2021) for more insights into the theory and application of normalization of FTMS for complex mixtures.."*

Lastly, there are a few different terms that we come across: intensity, magnitude, peak height. They are in my mind the same, but are there any nomenclature issues that need to be discussed in this regard? Should we stick to one uniform term for more comparable literature or keep using all of them interchangeably? Are there different connotations/flavors to these terms? Some ideas for talking points if you choose to include a terminology discussion (I do recommend! This paper would be the perfect spot for it).

**We agree that consistent terminology is helpful within and across publications. We did a search across our manuscript to ensure we are using 'intensity' and not other related terms. We made a couple small adjustments to keep consistent terminology. We note that 'magnitude' may be ambiguous given different signal processing types (e.g. absorption or magnitude mode FT processing). 'Peak height' may be acceptable, and indeed can be helpful to avoid ambiguity if intensity is an area or height derived metric. However, intensity is commonly used, colloquially, and is the output column name from Vendor provided FTICR peak picking tools (e.g. Bruker DataAnalysis indicates column 'I' for peak intensity). Of course, we recommend that researchers define the terminology used in their papers where ambiguity or uncertainty may exist.**

**We added the following text to the start of Section 2 to clarify our suggested approach:**

**"Further, we suggest consistent use of the term 'intensity' in FTMS NOM studies to describe how much signal is observed for a given peak, as opposed to 'height', 'magnitude', or other alternatives. While terminology is not our central focus, it is useful to pursue consistency across studies."**

#######

Reviewer 2:

General comments:

This study mainly showed that the peak intensity of FTICRMS is not a proxy for abundance and concentration. However, this is somehow common sense and should be already known to most researchers. The experiments performed by the authors nicely proved that peak intensity should be used with caution, especially for quantification. Again it is already well known.

**We agree that such information should be well known and basic knowledge to experienced mass spectrometrists, however many users of FTMS data are not formally trained in mass spectrometry. This manuscript serves as an accessible education on these pitfalls. Further, there are many papers that use peak intensities in ways that**

**implicitly assume the peak intensity is a proxy for abundance. More importantly, a key message from our paper is that despite the fact that peak intensities are not a direct proxy of abundance, the peak intensity data carries enough valid information to be useful. This is an important point that is not obvious and is not, to our knowledge, in the literature. It is also of practical value to the entire research community using FTMS to study NOM.**

The author claimed that there are plenty of misuse studies of peak intensity, if this is true, I would suggest the author organize a table of literature to illustrate how serious this issue is.

**Within the manuscript, we do not make this claim. Within a previous response to a reviewer, we remarked that we do observe some manuscripts now using peak intensities wherein about a decade ago such use was less common. It is not our goal to highlight specific papers and critique them, but to raise awareness of how to most robustly use peak intensities.**

The simulation model is great but I highly doubt its applicability considering so many speculation factors were introduced, including random error, etc. Besides, this model was to evaluate the matrix effect on peak intensity, and seems that it can only be used as an educational tool for people to understand the bias of FTRCRMS in the quantitative study. How people can use it for their own samples?

**As with all useful models, simplifications are necessary. The simplifications do not undermine the value of the model as it is, and they provide guidance for future development of the model (e.g., inclusion of non-random error).**

**In terms of how people can use it for their own samples, we provide guidance on this within Section 6. While we expect more sophisticated versions of the model will be developed in future work, the starting point of using it for real sample sets is setting the simulations to use the same number of peaks observed in each real sample or each pair of real samples. The sample-level peak intensity distribution could also be constrained to be similar between the simulation model and real samples.**

For the model part, I suggest the author simplify the sentences as it is hard for me to understand them and, if necessary, list the mathematical equations used for calculation. Add text to clearly and concisely describe what this simulation model is, how it can be applied to environmental samples, and how people can make use of it. This is very important because the model is the only highlight I can see from the manuscript even though it is very hard for me to understand all. I suggest a major revision for this manuscript and the author should pay attention to the concise of writing since the authors aim at people who are new in this field.

**We cut the simulation model section (section 5) by more than half to help with readability and broad accessibility. As part of revising section 5, we replaced most of the simulation**

**model description with a shorter and more broadly accessible summary of what the model does.**

For specific comments see below please:

The introduction part mainly described the diversity studies (yes or no question), then all of a sudden in the last paragraph, raised the concern of "quantification" (how much question), What is your point here? Could the diversity part be removed and more focused on how people used the peak intensity data incorrectly to support the concern of the authors?

**We edited the introduction to include more emphasis on the mean trait (or property) values. Those were in the previous version, but not emphasized. Our intention across the introduction to present the ecological metrics as continuous variables, and did not intend to present them as answering a yes or no question. We carefully read over the introduction to ensure the text does not convey a binary yes/no perspective.**

The study lacks a "Method" part, including information on compounds, standards, instruments, etc. Please provide. Sections 3.1 and 3.2 should be improved, use a scatter plot for Fig.4 instead so that you can simplify the text and make it easy for readers.

**The Methods are detailed in the Supplementary Information file, and include information on chemicals, sample preparation, mass spectrometry measurements, mass spectrometry data analysis and the simulation model. We have opted to keep this bulk of text within the SI as it would unnecessarily lengthen the main text of the manuscript. We added a sentence to Section 3 to indicate the location of the Methods text.**
***"The experimental methods used are described in detail in the Supplementary Information."***

It seems that the homogeneity of the sample sets is not taken into consideration. Real-world within peak comparison might be more complicated if the studied set contains very heterogeneous samples.

**The complexity of our empirical data increases from pure compounds in clean solvent (an ideal case), through to mixtures of chemicals in complex matrices (e.g. with organic matter added or artificial river water SPE eluent). While our sample set does not approach the heterogeneity of a real-world sample set, we show that even in the simple, idealized cases, peak intensities are not directly quantitative. Thus, in real-world studies with increased heterogeneity, this issue is only exacerbated.**

**We added a sentence to the penultimate paragraph of section 3 to this effect;**
***"A 'real-world' sample set would have even greater diversity and heterogeneity than presented here, and thus the issues with use of peak intensities for quantitative interpretation would only be exacerbated."***

L75: There was only one paper cited, but later sentence uses "these studies", add more papers please.

**Two additional citations have been added.**

L113: "Using consistent sample concentrations" is impossible

**We respectfully disagree here - in a typical NOM workflow it is readily possible to measure the TOC of the extract or to measure the mass of the dried extract. Several groups normalize the concentration (TOC) of their extracts prior to SPE, or prior to mass spectrometry measurement. Of course, frequently this is also not done, and variability in concentrations does exist.**

L145-L146: Add text to describe what samples, what compounds, and what chemical standards.

**We have clarified that the details are in the supplementary information. The specific chemicals are also detailed in the context of the Figure.**

L268-L269, list the publications that misuse peak intensities

**We do not think it is necessary or helpful to explicitly call out papers for misusing peak intensities - that is not our intention or inference. Our manuscript describes why peak intensity usage should be done carefully and the caveats and considerations which must be taken into account.**

L369: add literatures

**The text was edited and references were added. It now reads: "There is significant value in using FTMS data to study NOM chemistry (Bahureksa et al., 2021; Cooper et al., 2022; Spencer et al., 2015; Stubbins et al., 2010), and it is vital that this be done based on rigorous use of the data."**

L275-L276, why do you choose 100 and 1000 peaks? Where are these peaks from? From what kind of samples?

**These peak numbers represent a very sparse sample (100 peaks), and a more typical order of magnitude (1000) for FTMS NOM data acquired over the past decade. Newer and higher performing instruments can increase the number of detected features even more (10,000 or above), however our models did not significantly change when we evaluated more (10,000) peaks. We note that the law of large numbers, which may be a key factor in the model's observations, would suggest that results will only more closely reflect true properties as more peaks are added. Thus, the use of 100 and 1000 reflects more conservative scenarios.**

Figures: Be consistent, either use figure and Fig, do not mix the usage

**We edited text to make the usage consistent. We use (Fig. X) in parenthesis to indicate the referred to object, and 'figure(s)' in a sentence when the subject of the sentence is the figure.**

Fig.4 Please change it to a scatter plot and give a correlation value (R2), this can help people understand the relationship between intensity and concentration.

**We replaced the figure with a scatter plot. The Pearson r (not R2) correlation coefficients and p-values (calculated by the Python scipy stats module) have been calculated for each dataset and are included in a new supplementary table (Table S1). Figure 4 caption has been updated to reflect this.**

In fact, in lines 162-163, the author also suggested a calibration curve, why not do it for this study?

**As discussed in the text, comparisons of the same peak between samples are problematic because the same m/z (and same molecular formula) may be different isomers (or different relative amounts of the same isomers) between samples. As demonstrated in the empirical section, isomeric compounds can have starkly different ionization efficiency. Thus, a calibration curve would not resolve the case of NOM studies in which isomeric composition is unknown.**

**Our intention is to say: A calibration curve *could* be used in the case of a sample where you do know the specific chemical identity and can purchase a standard, but such efforts would require additional upfront separation (e.g., online liquid chromatography or ion mobility) to ensure the chemical you are calibrating is the analyte of interest. Such workflows would be extremely laborious, targeted, and sample-set specific.**

What is relative intensity?

**Relative intensity is a mass spectrometry term defined as the ratio of a signal of interest to the base peak (most intense peak in the spectrum). In Figure 4, we use a slightly modified definition to account for scaling across spectra, and this is detailed in the caption to Figure 4.**

Fig. 8: How do you get the observed difference? How is the true difference calculated?

**This is all done within the simulation model, and the text in Section 5 was edited to help make this clearer. The model generates a sample *in silico*, which is used as the true sample with zero error. The model then adds specific kinds of errors (as detailed in the manuscript) meant to represent the kind of error that occurs when running a real sample.**

**In this case we have both truth (i.e., no error) and observed (i.e., after error is introduced to the truth).**

The R2 in Figure 8C is questionable: although the value appears to be high, it's obvious that the independent variables are heteroscedastic.

**This figure and its companion (based on a sensitivity analysis) are now both in the Supplementary Material and are Figures S6 and S7. To highlight the issue with heteroscedasticity, we added the following text to the Supplementary Methods (Section 2.6): "However, we suggest caution when interpreting the R2 values associated with Figure S6C and S7C as the differences collapse when near zero, leading to heteroscedastic residuals that likely bias the R2." We also added the following text to the captions of Figures S6 and S7: "On panel C the R2 value should be interpreted with caution as the residuals are clearly heteroscedastic."**

---

## Author Response (AR3)

Dear Dr. Middelburg,

Thank you for helping our manuscript continue to improve.

Your requested edits have been completed.

Thank you,

James Stegen (on behalf of all co-authors)